# The VP53 protein encoded by RNA2 of a fabavirus, broad bean wilt virus 2, is essential for viral systemic infection
Myung-Hwi Kim[1,5], Boram Choi[2,5], Seok-Yeong Jang[3], Ji-Soo Choi[3], Sora Kim[3], Yubin Lee[3], Suejin Park[4], Sun-Jung Kwon[2], Jin-Ho Kang[2,3] & Jang-Kyun Seo [1,2,3] ✉

Plant viruses evolves diverse strategies to overcome the limitations of their genomic capacity and express multiple proteins, despite the constraints imposed by the host translation system. Broad bean wilt virus 2 (BBWV2) is a widespread viral pathogen, causing severe damage to economically important crops. It is hypothesized that BBWV2 RNA2 possesses two alternative in-frame translation initiation codons, resulting in the production of two largely overlapping proteins, VP53 and VP37. In this study, we aim to investigate the expression and function of VP53, an N-terminally 128-amino-acid-extended form of the viral movement protein VP37, during BBWV2 infection. By engineering various recombinant and mutant constructs of BBWV2 RNA2, here we demonstrate that VP53 is indeed expressed during BBWV2 infection. We also provide evidence of the translation of the two overlapping proteins through ribosomal leaky scanning. Furthermore, our study highlights the indispensability of VP53 for successful systemic infection of BBWV2, as its removal results in the loss of virus infectivity. These insights into the translation mechanism and functional role of VP53 during BBWV2 infection significantly contribute to our understanding of the infection mechanisms employed by fabaviruses.

Plant viruses are obligate parasitic pathogens that rely on living cells and host machineries for their replication and propagation. To express their proteins, plant viruses are entirely dependent on the host eukaryotic translation system, which typically translates only the first open reading frame (ORF) in an mRNA[1]. This dependence poses a significant challenge for plant viruses, as most of them have compact genomes consisting of a single or a few RNA or DNA molecules encoding multiple viral proteins[2]. Consequently, to overcome this limitation and express multiple proteins, viruses have evolved diverse strategies, such as genome segmentation, subgenomic RNA production, polyprotein processing, ribosomal leaky scanning, internal ribosome entry, ribosomal frameshifting, stop codon read-through, etc.[1,3–5].

Broad bean wilt virus 2 (BBWV2) is a widely distributed viral pathogen that exhibits a broad host range and poses a significant threat to numerous economically important crops, including various legume species[6]. In recent years, BBWV2 has emerged as a prevalent virus, displaying increasing incidence and severity in several horticultural crops in Korea, such as pepper, Chinese yam, and sesame[7–9]. Extensive studies have identified numerous strains and isolates of BBWV2 infecting various plant species, enabling the determination of their genome sequences[8,10,11]. Population genetic analyses conducted on these BBWV2 strains and isolates have provided insights into the high evolutionary plasticity of the BBWV2 genome, contributing to genetic and pathogenic diversification[8,12].

BBWV2 belongs to the genus *Fabavirus* in the family *Secoviridae*, which comprises 106 virus species[13]. The genome of the members in the family *Secoviridae* consists of one or two linear positive-sense RNA molecules that encode polyproteins. Upon translation, these polyproteins are subsequently processed into functional mature proteins[13]. The BBWV2 genome is composed of two single-stranded RNA molecules, RNA1 and RNA2, which are ~5960 and 3600 nucleotides (nt) in length, respectively[6]. BBWV2 RNA1 contains a single ORF that encodes a single polyprotein precursor. This precursor undergoes proteolytic cleavage to yield five mature proteins: protease cofactor (Co-Pro), NTP-binding motif, viral genome-linked protein, protease (Pro), and RNA-dependent RNA polymerase (Fig. 1a). Similar to other related viruses in the family *Secoviridae*, such as cowpea mosaic virus (CPMV) and bean pod mottle virus

[1]Department of Agricultural Biotechnology, Seoul National University, Seoul 08826, Republic of Korea. [2]Institutes of Green Bio Science and Technology, Seoul National University, Pyeongchang 25354, Republic of Korea. [3]Department of International Agricultural Technology, Seoul National University, Pyeongchang 25354, Republic of Korea. [4]Department of Horticulture, Jeonbuk National University, Jeonju 54896, Republic of Korea. [5]These authors contributed equally: Myung-Hwi Kim, Boram Choi. ✉e-mail: jangseo@snu.ac.kr

**Fig. 1 | GFP tagging of BBWV2 RNA2 at the C-terminus of VP53/VP37. a** The genome structure of broad bean wilt virus 2 (BBWV2). BBWV2 genome consists of two single-stranded RNA molecules, RNA1 and RNA2. RNA1 contains a single ORF that encodes a single polyprotein precursor, which is cleaved to yield five mature proteins: protease cofactor (Co-Pro), NTP-binding motif (NTBM), viral genome-linked protein (VPg), protease (Pro), and RNA-dependent RNA polymerase (RdRp). RNA2 is assumed to be translated into two largely overlapping polyproteins using two alternative in-frame initiation codons (at nucleotide positions 236 and 620). These two polyproteins are processed twice at the same cleavage sites, finally yielding four mature proteins: VP53, VP37, large coat protein (LCP), and small coat protein (SCP). The proteolytic cleavage sites recognized by Pro are indicated by arrows. **b** Schematic representation of a cDNA clone of BBWV2 RNA2 tagged with GFP (pBBWV2-R2-53/37-GFP). pBBWV2-R2-53/37-GFP contains the VP53 and VP37 cistrons fused in-frame with GFP, thereby expressing VP53-GFP and VP37-GFP, respectively, during virus replication. **c** Virulence of cDNA clones of wild-type (wt) BBWV2 (pBBWV2-RP1-R1 + pBBWV2-RP1-R2) and BBWV2-53/37-GFP (pBBWV2-RP1-R1 + pBBWV2-R2-53/37-GFP) in *Nicotiana benthamiana*. *N. benthamiana* plants inoculated with wt BBWV2 or BBWV2-53/37-GFP were observed using a FOBI fluorescence imaging system at 6 dpi. **d** Confocal microscopic observation of leaf tissues infected with either wt BBWV2 or BBWV2-53/37-GFP. Callose stained with aniline blue was utilized as a marker for plasmodesmata (PD). Aniline blue-stained callose and mitochondria emit cyan and red fluorescence, respectively. Bar = 10 μm. Data shown are representatives of at least three independent experiments.

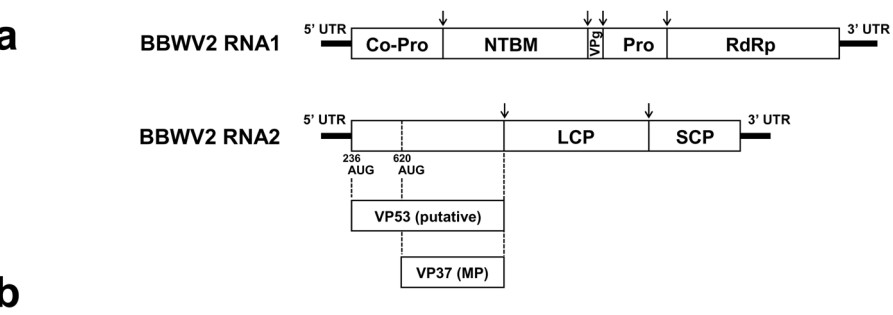

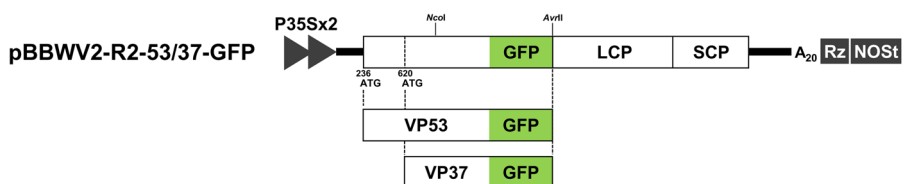

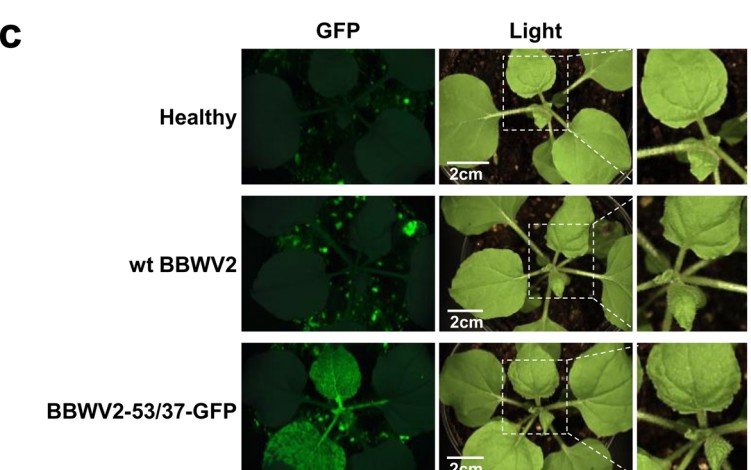

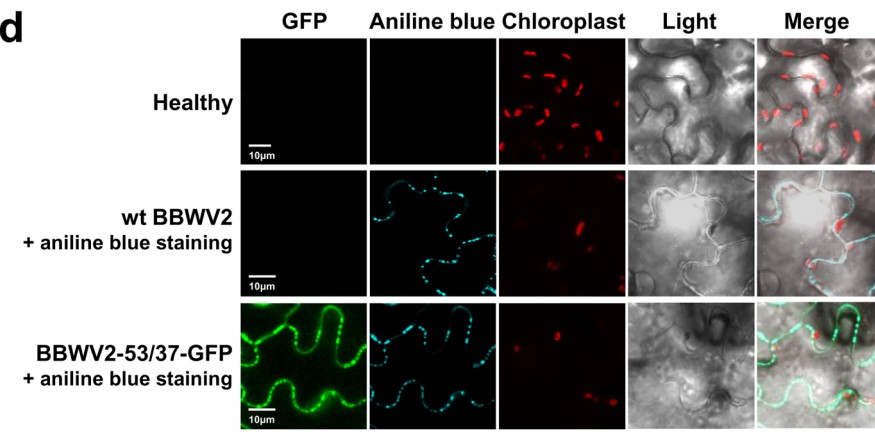

(BPMV)[14–16], BBWV2 RNA2 is hypothesized to encode two largely overlapping polyproteins using two alternative in-frame initiation codons (at nucleotide positions 236 and 620); however, this has not yet been demonstrated experimentally (Fig. 1a). These two polyproteins, which differ only at their N-termini, are processed twice at the same cleavage sites, finally yielding four mature proteins: the large and small coat proteins (LCP and SCP, respectively), a putative larger N-terminal protein of 53 kDa, designated as VP53 (with an unknown function), and the smaller N-terminal protein of 37 kDa, designated as VP37, which functions as the viral movement protein (Fig. 1a).

In vivo expression and function of VP37 have been demonstrated in previous studies[17–19]. VP37 has been shown to form tubules in the plasmodesmata (PD) that mediate the tubule-guided cell-to-cell movement of BBWV2 virions[17,18]. In addition, VP37 has the ability to bind single-strand

nucleic acids and suppress RNA silencing [20,21]. However, it has not been demonstrated whether VP53, the N-terminally 128-amino-acid-extended form of VP37, is truly expressed during viral replication and required for successful infection of BBWV2. In our previous studies, we generated infectious cDNA clones of BBWV2 RNA1 and RNA2 using a T-DNA-based binary vector that can be delivered into plant cells via Agrobacterium-mediated inoculation (agroinfiltration)[6,22,23]. In the present study, utilizing this BBWV2 infectious cDNA clone system, we generated various recombinant and mutant constructs of BBWV2 RNA2 to examine whether VP53 is expressed from RNA2 and necessary for BBWV2 infection. Our results showed that VP53 was detectably expressed during BBWV2 infection, although its accumulation level was significantly lower than that of VP37. Furthermore, the removal of the N-terminal non-overlapping region of VP53 from RNA2 resulted in the loss of infectivity of the virus. Together, our results suggest that VP53 is essential for BBWV2 infectivity.

## Results

### GFP tagging of BBWV2 RNA2 at the C-terminus of VP53/VP37 does not impair virus infectivity

We first sought to examine whether tagging at the C-termini of VP53 and VP37 affects virus infectivity. To this end, we generated a BBWV2 RNA2 construct tagged with GFP by engineering the C-terminal region of VP53/VP37 of pBBWV2-R2-OE (Fig. 1b)[23]. The engineered construct, named pBBWV2-R2-53/37-GFP, contained the VP53 and VP37 cistrons fused in-frame with GFP, thereby expressing VP53-GFP and VP37-GFP, respectively, during virus replication. To evaluate the infectivity and GFP expression of pBBWV2-R2-53/37-GFP, we inoculated *N. benthamiana* plants with a mixture of *Agrobacterium* cultures containing pBBWV2-RP1-R1 and pBBWV2-R2-53/37-GFP (this combination was designated as BBWV2-53/37-GFP). *N. benthamiana* plants inoculated with BBWV2-53/37-GFP exhibited vein chlorosis and leaf malformation symptoms in systemic leaves, whereas wild-type (wt) BBWV2 (pBBWV2-RP1-R1 + pBBWV2-RP1-R2) induced very mild symptoms (Fig. 1c), although no significant difference in viral RNA accumulation was observed between wt BBWV2 and BBWV2-53/37-GFP (Supplementary Fig. S1). Inoculated plants were observed using a FOBI fluorescence imaging system at 6 dpi. Strong GFP signals were observed in the systemic leaves of the plants inoculated with BBWV2-53/37-GFP (Fig. 1c). RT-PCR detection confirmed that the plants inoculated with BBWV2-53/37-GFP were systemically infected. Our results revealed that GFP tagging at the C-terminus of VP53/VP37 did not significantly impair the infectivity of BBWV2 and remained stable during virus replication.

### Subcellular distribution of BBWV2 VP37 and VP53 in plant cells

VP37 is likely to localize in the PD because VP37 forms tubule structures in the PD[17,18]. However, the subcellular distribution of VP37 and VP53 in the BBWV2-infected cells has not yet been examined. Thus, to address this, we observed the leaf tissues infected with BBWV2-53/37-GFP using confocal microscopy. The GFP signals were observed as punctate spots along the cell periphery, reminiscent of PD localization (Fig. 1d). These spots co-localized with aniline blue-stained callose, which serves as a PD marker, indicating that VP37-GFP and/or VP53-GFP expressed from BBWV2-53/37-GFP localize to the PD (Fig. 1d). However, this approach was unable to differentiate the subcellular localization of VP37 and VP53 because both proteins were expressed as GFP-fusion proteins in the cells infected with BBWV2-53/37-GFP. To overcome this limitation, we utilized an alternative approach wherein VP37-GFP and VP53-GFP were individually expressed in the leaves of *N. benthamiana* plants using an *Agrobacterium*-mediated transient expression system[24]. Results are summarized in Fig. 2. Notably, VP37-GFP predominantly accumulated in the PD, as evidenced by its colocalization with aniline blue-stained callose. In contrast, VP53-GFP was observed as large inclusions in the cytoplasm, without showing specific localization in the PD. The distinct subcellular localization of VP37 and VP53 was particularly intriguing because the C-terminal three-fourths of VP53 correspond to VP37. Based on these findings, we concluded that the GFP signals

observed in the PD of the cells infected with BBWV2-53/37-GFP were specifically attributed to VP37-GFP (Fig. 1d).

### VP53 is expressed during BBWV2 infection

Since GFP tagging of BBWV2 RNA2 at the C-terminus of VP53/VP37 did not impair virus infectivity (Fig. 1), we generated another BBWV2 RNA2 construct tagged with a Flag epitope at the C-terminus of VP53/VP37 (Fig. 3a). The engineered construct, named pBBWV2-R2-53/37-Flag, was expected to express VP53-Flag and VP37-Flag during virus replication (Fig. 3a). To evaluate the infectivity of pBBWV2-R2-53/37-Flag and the expression of VP53-Flag and VP37-Flag, we inoculated *N. benthamiana* plants with a mixture of *Agrobacterium* cultures containing pBBWV2-RP1-R1 and pBBWV2-R2-53/37-Flag (this combination was designated as BBWV2-53/37-Flag). BBWV2-53/37-Flag induced similar mild symptoms in *N. benthamiana* plants as wt BBWV2, indicating that BBWV2-53/37-Flag is fully infectious and has the same virulence as wt BBWV2.

We next examined the accumulation levels of VP53-Flag and VP37-Flag during virus infection. Total proteins were extracted from the symptomatic leaves of *N. benthamiana* plants infected with BBWV2-53/37-Flag and analyzed by Western blotting using anti-Flag antibodies. Our analysis clearly detected both VP53-Flag and VP37-Flag at the expected size (Fig. 3b). VP53-Flag accumulated to a barely detectable level, while the accumulation level of VP37-Flag was much higher than that of VP53-Flag (Fig. 3b). Furthermore, a gel fragment corresponding to the molecular weight of VP53-Flag (~53 kDa) was excised from the SDS-PAGE gel, that separated the total protein extracted from *N. benthamiana* leaves infected with BBWV2-53/37-Flag. Subsequently, the excised gel fragment was subjected to in-gel digestion using trypsin followed by LC-MS/MS analysis. The analysis identified several peptide sequences corresponding to the N-terminal VP53 sequence, which does not overlap with VP37 (Supplementary Fig. S2). These results demonstrated that both VP53 and VP37 were translated from BBWV2 RNA2 and raised the possibility that the AUG codons for VP53 and VP37 may differ in their translation initiation efficiency. However, we cannot rule out other possibilities, such as differences in protein folding and accessibility to the Flag epitope, or variations in protein stability between VP53 and VP37.

Because the accumulation levels of VP53 and VP37 were significantly different, we investigated the sequence contexts of the start codons for VP53 and VP37. Our analyses revealed that the nucleotide sequences around the start codon for VP37 were highly conserved among BBWV2 isolates, whereas some variations existed for VP53. (Fig. 3c, d); however, no significant conservation was observed among fabaviruses and comoviruses (Supplementary Fig. S3). Furthermore, the contexts around both the VP53 and VP37 start codons [G(U/C)UGUAAUGCG and GAAAUCAUGA(G/A), respectively] were predicted to be unfavorable for eukaryotic ribosome recognition [cf. the consensus AUG context in higher plants: aA(A/C)aAUGGC] (Fig. 3c, d)[25,26].

The efficiency of a certain start codon can be influenced by the RNA secondary structures located upstream and downstream of the start codon[27,28]. Therefore, we performed RNA structure analyses for the sequences surrounding the VP53 and VP37 start codons. Our analyses revealed that the VP53 start codon was located within a strong stem-loop RNA structure (Fig. 3e). In contrast, weak stem-loop RNA structures were predicted immediately upstream and downstream of the VP53 start codon (Fig. 3f).

### VP53 is required for systemic infection of BBWV2

To examine the requirement and function of VP53 and VP37 in BBWV2 infection, we generated BBWV2 RNA2 mutant constructs in which specific nucleotide mutations were introduced into the initiation codon for VP53 or VP37, leading to the disruption of their expression. As depicted in Fig. 4a, pBBWV2-R2-53/37$^{A620G}$-GFP contained an A-to-G substitution at nucleotide position 620, resulting in the loss of the ATG initiation codon for VP37. Consequently, the virus would be incapable of expressing VP37, while still retaining the capability to express VP53, except for a single amino

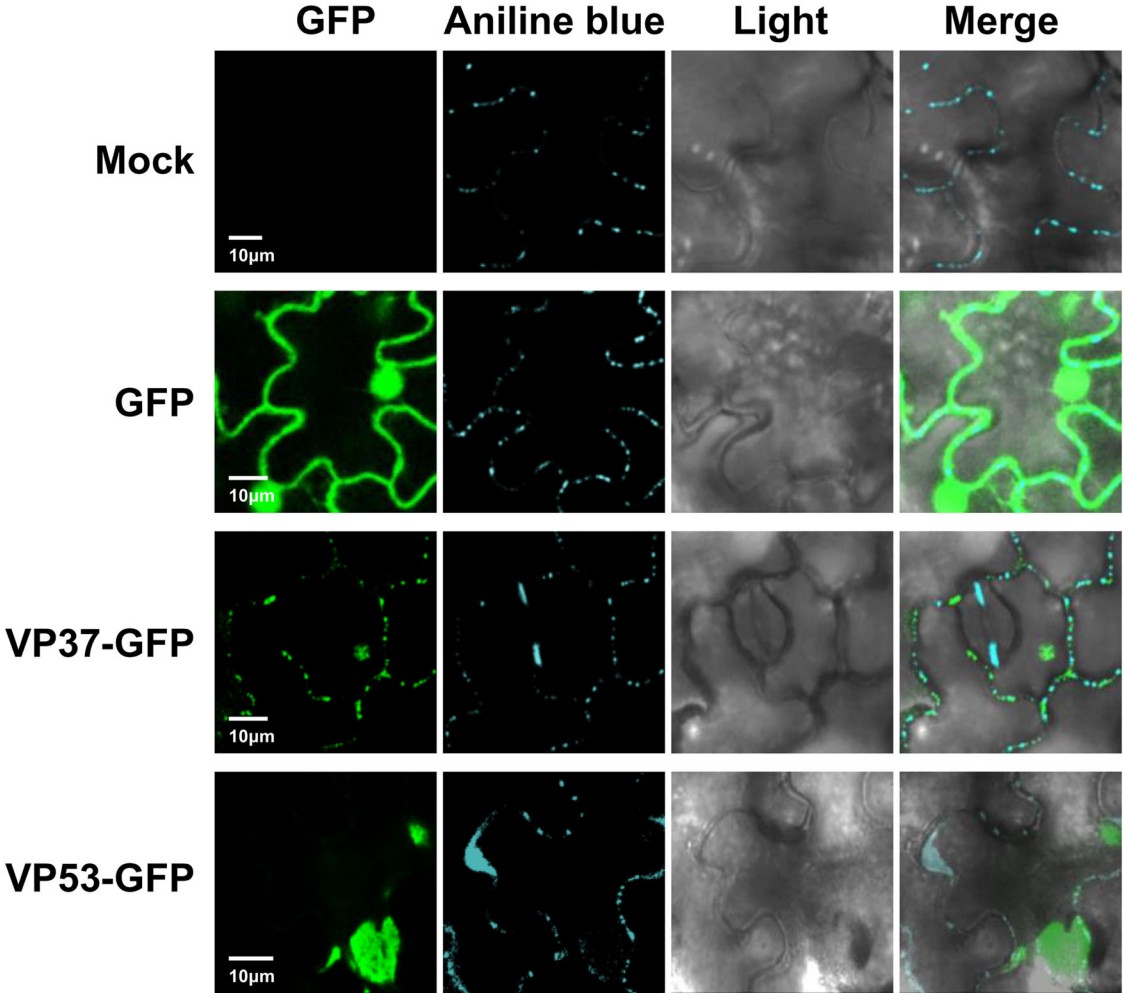

**Fig. 2 | Subcellular distribution of VP37-GFP and VP53-GFP in plant cells.** *Agrobacterium* transformants expressing either GFP, VP37-GFP, or VP53-GFP were infiltrated into leaves of *N. benthamiana* plants. At 3 dpi, the infiltrated leaf tissues were stained with aniline blue to visualize callose as a PD marker, and then analyzed using confocal microscopy. Bar = 10 μm.

acid substitution [methionine (ATG) to valine (GTG)] at amino acid position 129. In contrast, pBBWV2-R2-53$^{G238A}$/37-GFP and pBBWV2-R2-53$^{ATG236TAA}$/37-GFP contained mutations at nucleotide positions 238 and 236–238, respectively (Fig. 4a). The former had a G-to-A substitution at position 238, while the latter had an ATG-to-TAA substitution at positions 236–238. Thus, these viruses lost the ATG initiation codon for VP53, rendering them capable of expressing VP37 but not VP53.

*N. benthamiana* plants were agroinfiltrated with pBBWV2-RP1-R1 and either pBBWV2-R2-53/37$^{A620G}$-GFP, pBBWV2-R2-53$^{G238A}$/37-GFP, or pBBWV2-R2-53$^{ATG236TAA}$/37-GFP (these combinations were designated as BBWV2-53/37$^{A620G}$-GFP, BBWV2-53$^{G238A}$/37-GFP, and BBWV2-53$^{ATG236TAA}$/37-GFP, respectively). To assess the infectivity of the mutant viruses, the inoculated plants were examined using a FOBI fluorescence imaging system at 8 dpi. While BBWV2-53/37-GFP successfully established systemic infection, BBWV2-53/37$^{A620G}$-GFP was unable to infect plants (Fig. 4b), indicating that VP37 is essential for systemic infection of BBWV2. In contrast, out of the 12 plants inoculated with BBWV2-53$^{G238A}$/37-GFP, ten plants exhibited systemic infection. In addition, five out of the 12 plants inoculated with BBWV2-53$^{ATG236TAA}$/37-GFP displayed systemic infection. Virus infection of the inoculated plants was verified through RT-PCR detection (Supplementary Fig. S4). To evaluate the replication competence of the BBWV2 RNA2 mutants, we performed strand-specific RT-qPCR analysis to detect replication-specific (-)-strand RNAs. Total RNA was isolated from the leaves infiltrated with BBWV2-53/37-GFP, BBWV2-53/37$^{A620G}$-GFP, BBWV2-53$^{G238A}$/37-GFP, or BBWV2-53$^{ATG236TAA}$/37-GFP at 2

dpi and analyzed by strand-specific RT-qPCR. Although BBWV2-53/37$^{A620G}$-GFP failed to establish systemic infection, this mutant accumulated a significant amount of (-)-strand RNA1 and RNA2 during the early replication phase (Fig. 4c, d), indicating that VP37 is indispensable for systemic infection. In contrast, BBWV2-53$^{G238A}$/37-GFP and BBWV2-53$^{ATG236TAA}$/37-GFP exhibited a relatively lower accumulation of (-)-strand RNA1 and RNA2 compared to BBWV2-53/37-GFP (Fig. 4c, d), suggesting that VP53 may function in BBWV2 replication. These unexpected results prompt us to suspect that the introduced mutations were insufficient to disrupt the expression of VP53, and that there might be minimal translation at the non-AUG codons, such as GUG and AUA.

We next sought to examine whether the nucleotide mutations in the initiation codon for VP53 and VP37 effectively led to the impairment of VP53 and VP37 expression, respectively. To this end, we engineered pBBWV2-R2-53/37-Flag to introduce either an A-to-G substitution at nucleotide position 620 or an ATG-to-TAA substitution at positions 236-238 (Fig. 5a). The resulting mutant constructs, named pBBWV2-R2-53/37$^{A620G}$-Flag and pBBWV2-R2-53$^{ATG236TAA}$/37-Flag, were agroinfiltrated along with pBBWV2-RP1-R1 into *N. benthamiana* plants (these combinations were designated as BBWV2-53/37$^{A620G}$-Flag and BBWV2-53$^{ATG236TAA}$/37-Flag). Virus infection of the inoculated plants was determined by observing symptom development on the systemic leaves (Fig. 5b) and confirmed by RT-PCR detection. Similar to the GFP-tagged BBWV2 mutants (Fig. 4), BBWV2-R2-53/37$^{A620G}$-Flag exhibited an inability to infect the plants, while 6 out of the 12 plants inoculated with BBWV2-53$^{ATG236TAA}$/

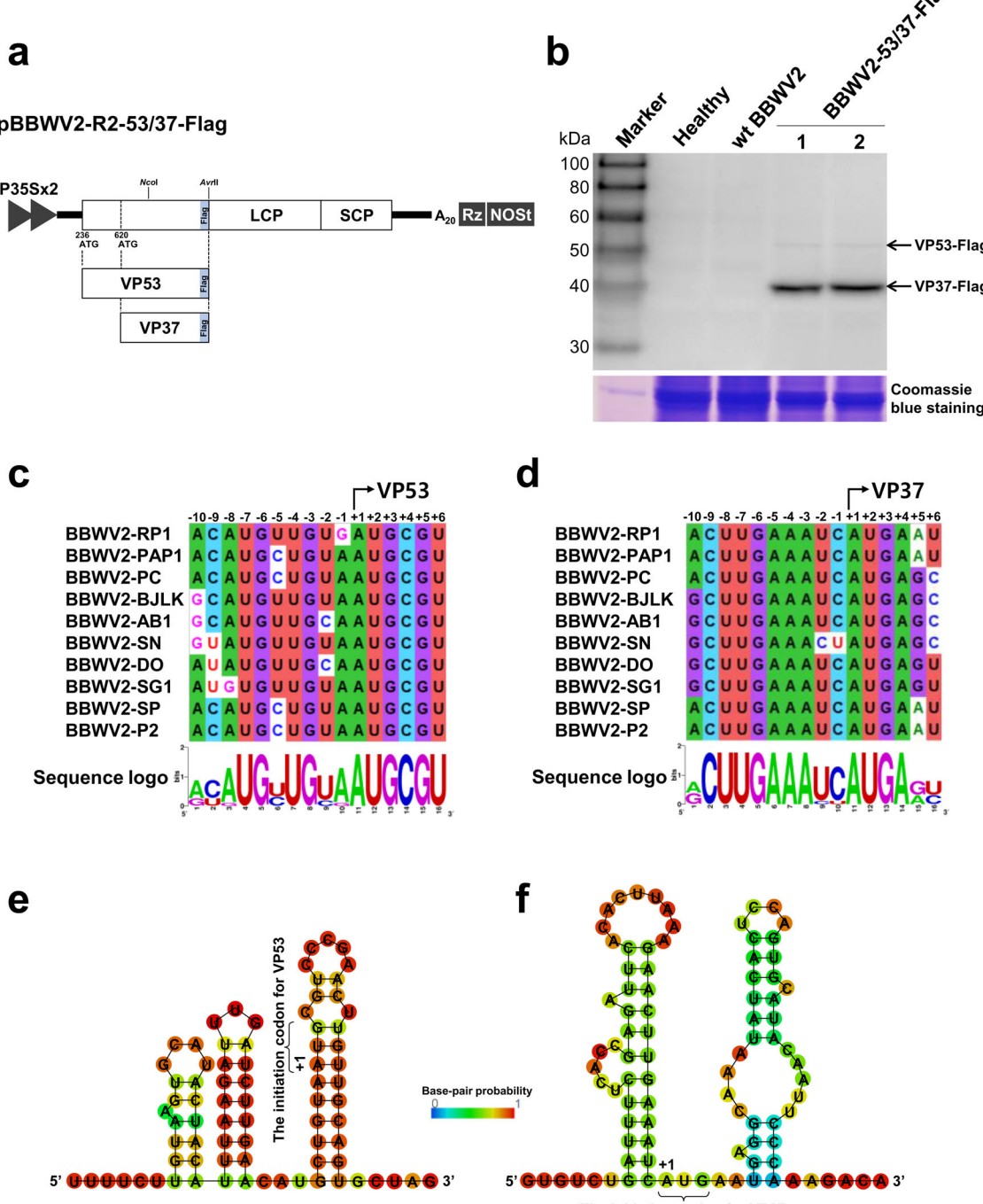

**Fig. 3 | Expression of VP53 and VP37 during BBWV2 infection. a** Schematic representation of a cDNA clone of BBWV2 RNA2 tagged with a Flag epitope (pBBWV2-R2-53/37-Flag). pBBWV2-R2-53/37-Flag contains the VP53 and VP37 cistrons fused in-frame with a Flag epitope, thereby expressing VP53-Flag and VP37-Flag, respectively, during virus replication. **b** Western blot analysis of the expression of VP53 and VP37. *N. benthamiana* plants were inoculated with wt BBWV2 or BBWV2-53/37-Flag (pBBWV2-RP1-R1 + pBBWV2-R2-53/37-GFP). Total protein was extracted from the upper symptomatic leaves of the inoculated plants and subjected to immunoblot analysis using anti-Flag antibodies. Protein size markers are indicated on the left side of the blot. A Coomassie blue-stained gel is shown below the blot as a loading control. The blot image has been cropped for clarity, and the original blot image is presented in Supplementary Fig. S6. Analysis of consensus sequences surrounding the AUG start codons for VP53 (**c**) and VP37 (**d**) among various BBWV2 isolates. Consensus sequences were analyzed using ClustalW and illustrated using WebLogo, which presented stacks of symbols. The size of symbols within the stack reflects the relative frequency of each base at that position. GenBank accession numbers: BBWV2-RP1 (KT380023), BBWV2-PAP1 (KT380021), BBWV2-PC (MW939477), BBWV2-BJLK (OP785726), BBWV2-AB1 (MH447989), BBWV2-SN (KX686590), BBWV2-DO (KT246496), BBWV2-SG1 (KJ789137), BBWV2-SP (KC625518), and BBWV2-P2 (KC625512). The stable hairpin RNA structures predicted within the RNA2 sequences surrounding the AUG start codons for VP53 (**e**) and VP37 (**f**). RNA secondary structures were predicted by RNAfold. The color code indicates base-pairing probabilities calculated with RNAfold.

**Fig. 4 | Effects of nucleotide substitution mutations introduced into the initiation codon for VP53 or VP37 on the virulence of BBWV2.**
**a** Schematic representation of cDNA clones of BBWV2 RNA2 mutant constructs. pBBWV2-R2-53/37^A620G^-GFP contained an A-to-G substitution at nucleotide position 620, resulting in the loss of the ATG initiation codon for VP37. pBBWV2-R2-53^G238A^/37-GFP and pBBWV2-R2-53^ATG236TAA^/37-GFP contained a G-to-A substitution at nucleotide position 238 and an ATG-to-TAA substitution at nucleotide positions 236-238, respectively, resulting in the loss of the ATG initiation codons for VP53.
**b** Virulence of BBWV2-53/37^A620G^-GFP, BBWV2-53^G238A^/37-GFP, and BBWV2-53^ATG236TAA^/37-GFP in *N. benthamiana*. *N. benthamiana* plants inoculated with either BBWV2-53/37-GFP, BBWV2-53/37^A620G^-GFP, BBWV2-53^G238A^/37-GFP, or BBWV2-53^ATG236TAA^/37-GFP were observed using a FOBI fluorescence imaging system at 8 dpi. Data shown are representatives of three independent experiments, with each experiment involving the inoculation of four plants per viral construct. In total, twelve plants were tested for the systemic infectivity of each virus (number of plants infected/number of plants inoculated). Virus infection of the inoculated plants was verified by RT-PCR detection. Relative accumulation levels of (-)-strand RNAs of BBWV2 mutant viruses. Total RNA was isolated from *N. benthamiana* leaves inoculated with either BBWV2-53/37-GFP, BBWV2-53/37^A620G^-GFP, BBWV2-53^G238A^/37-GFP, or BBWV2-53^ATG236TAA^/37-GFP at 2 dpi and subjected to strand-specific RT-qPCR to analyze the relative accumulation levels of (-)-strand RNA1 (**c**) and RNA2 (**d**). The mean ± SD of three replications is shown, and each column represents one group with nine plants. Significant differences indicated by different letters (*P* < 0.05) were analyzed using one-way ANOVA with Tukey's HSD test.

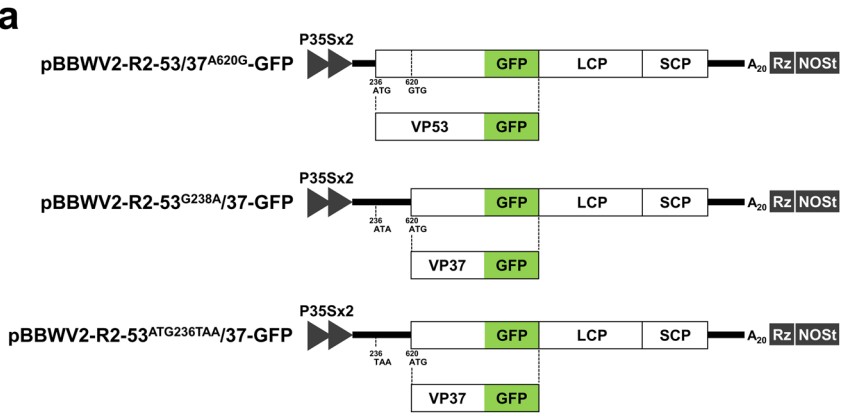

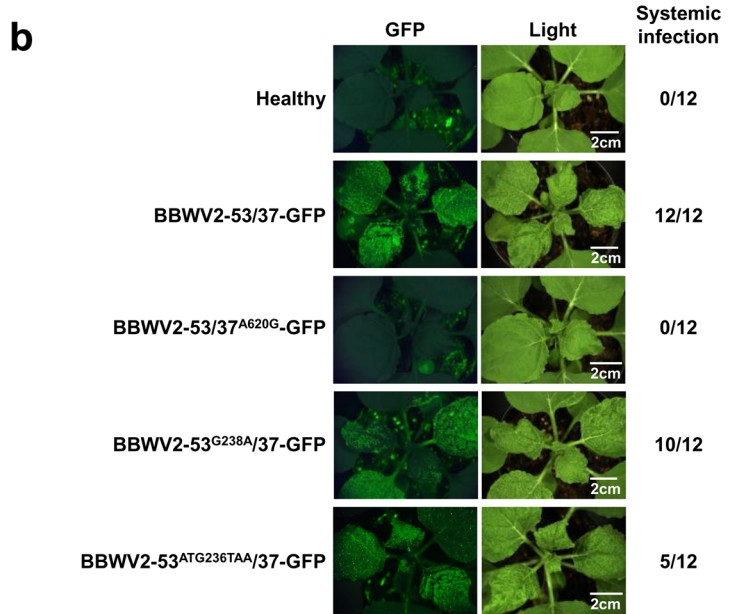

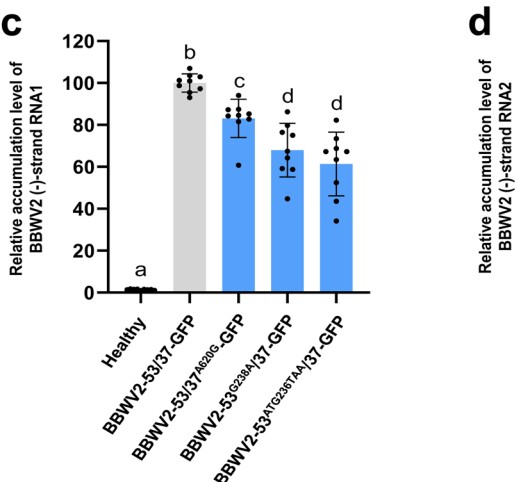

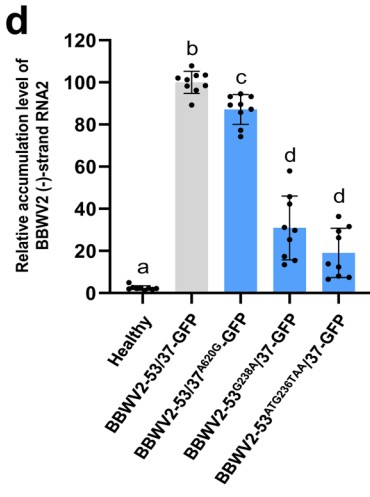

37-Flag exhibited systemic infection (Fig. 5b). To examine the expression of VP53-Flag and VP37-Flag from the mutant viruses, total proteins were extracted at 6 dpi from the infiltrated leaves of the tested plants, as well as the symptomatic upper leaves of *N. benthamiana* plants infected with BBWV2-53^ATG236TAA^/37-Flag. These protein samples were then subjected to Western blot analysis using an anti-Flag antibody. In the inoculated leaves with BBWV2-R2-53/37^A620G^-Flag, no accumulation of VP37-Flag was observed,

but a smaller protein band with an approximate size of 33 kDa was detected (Fig. 5c). This finding suggests that the A-to-G substitution at nucleotide position 620 effectively impaired VP37 expression (at least, translation at this non-AUG codon did not appear to occur at a detectable level), and that translation occurred at a next downstream ATG codon (probably at nucleotide positions 749-751). In contrast, the accumulation of VP53-Flag was detected in both the inoculated and upper systemic leaves of *N.*

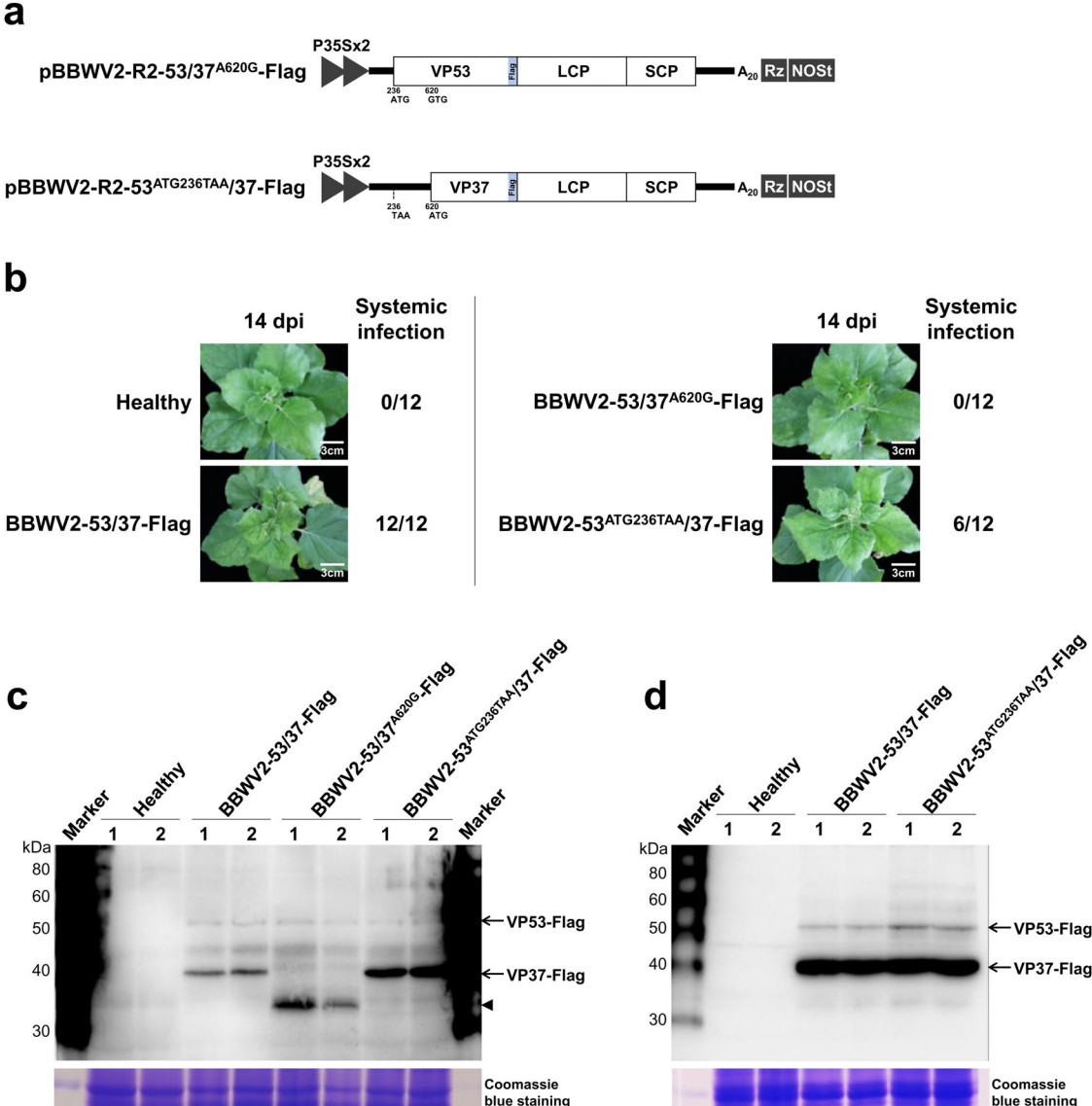

**Fig. 5 | Effects of nucleotide substitution mutations in the initiation codon for VP53 and VP37 on the expression of VP53 and VP37. a** Schematic representation of cDNA clones of the BBWV2 RNA2 constructs. pBBWV2-R2-53/37$^{A620G}$-Flag contains an A-to-G substitution at nucleotide position 620, resulting in the loss of the ATG initiation codon for VP37. pBBWV2-R2-53$^{ATG236TAA}$/37-Flag contains an ATG-to-TAA substitution at nucleotide positions 236-238, resulting in the loss of the ATG initiation codons for VP53. **b** Virulence of BBWV2-53/37$^{A620G}$-Flag and BBWV2-53$^{ATG236TAA}$/37-Flag in *N. benthamiana*. *N. benthamiana* plants were inoculated with either BBWV2-53/37-Flag, BBWV2-53/37$^{A620G}$-Flag, or BBWV2-53$^{ATG236TAA}$/37-Flag. Virus infection of the inoculated plants was determined by observing symptom development on the systemic leaves and confirmed by RT-PCR detection at 14 dpi. Data shown are representatives of three independent experiments, with each experiment involving the inoculation of four plants per viral construct. In total, 12 plants were tested for the systemic infectivity of each virus (number of plants infected/number of plants inoculated). Western blot analysis of the expression of VP53 and VP37 in the inoculated leaves (**c**) and upper symptomatic leaves (**d**). *N. benthamiana* plants were inoculated with either BBWV2-53/37-Flag, BBWV2-53/37$^{A620G}$-Flag, or BBWV2-53$^{ATG236TAA}$/37-Flag. Total protein was extracted at 6 dpi from the inoculated leaves and upper symptomatic leaves of the inoculated plants and subjected to immunoblot analysis using anti-Flag antibodies. Protein size markers are indicated on the left side of the blots. A Coomassie blue-stained gel is shown below each blot as a loading control. Arrowhead indicates the presence of a protein band (33 kDa) that is likely the result of translation initiation occurring at a downstream ATG (at nucleotide positions 749-751). The blot image has been cropped for clarity, and the original blot image is presented in Supplementary Fig. S6.

*benthamiana* plants infected with BBWV2-53$^{ATG236TAA}$/37-Flag (Fig. 5c, d). These results revealed that the mutations in the ATG initiation codon for VP53 did not effectively hinder the expression of VP53.

To determine the underlying reason for this finding, we examined whether the substitution mutations into the ATG initiation codon for VP53 are retained in the genomes of progeny viruses. The RNA2 genomic sequences of progeny viruses recovered by RT-PCR from the systemic leaves infected with BBWV2-53$^{G238A}$/37-GFP, BBWV2-53$^{ATG236TAA}$/37-GFP, or BBWV2-53$^{ATG236TAA}$/37-Flag were analyzed. The results showed that the G-to-A substitution at position 238 was restored to G in three individual progenies obtained from *N. benthamiana* plants infected with BBWV2-53$^{G238A}$/37-GFP (Fig. 6a). More interestingly, in all analyzed progenies that were obtained from *N. benthamiana* plants infected with BBWV2-53$^{ATG236TAA}$/37-GFP or BBWV2-53$^{ATG236TAA}$/37-Flag, the ATG-to-TAA substitution at positions 236-238 remained intact (Fig. 6b). However, a new G-to-A substitution at position 257 emerged in these progenies, leading to the creation of a novel ATG codon in-frame with VP53 (Fig. 6b). Based on these findings, it can be inferred that the translation initiation codon for VP53 has the potential to be restored or newly created through spontaneous mutations during virus replication.

**Fig. 6 | Analysis of the 5' genomic sequences of the progeny viruses recovered from the systemic leaves infected with BBWV2-53^G238A/37-GFP, BBWV2-53^ATG236TAA/37-GFP, or BBWV2-53^ATG236TAA/37-Flag.** The sequences of three independent progeny viruses were recovered by RT-PCR from three individual plants infected with each mutant virus. The nucleotide and deduced amino acid sequences of the recovered progeny viruses were aligned with those of the wt virus and the original nucleotide substitution mutants to highlight the positions and identities of the compensatory mutations. **a** The G-to-A substitution at position 238 was restored to G in all analyzed progenies. **b** The ATG-to-TAA substitution at positions 236-238 remained intact, while a new G-to-A substitution at position 257 emerged in all analyzed progenies. This substitution might result in the creation of a novel ATG codon in-frame with VP53.

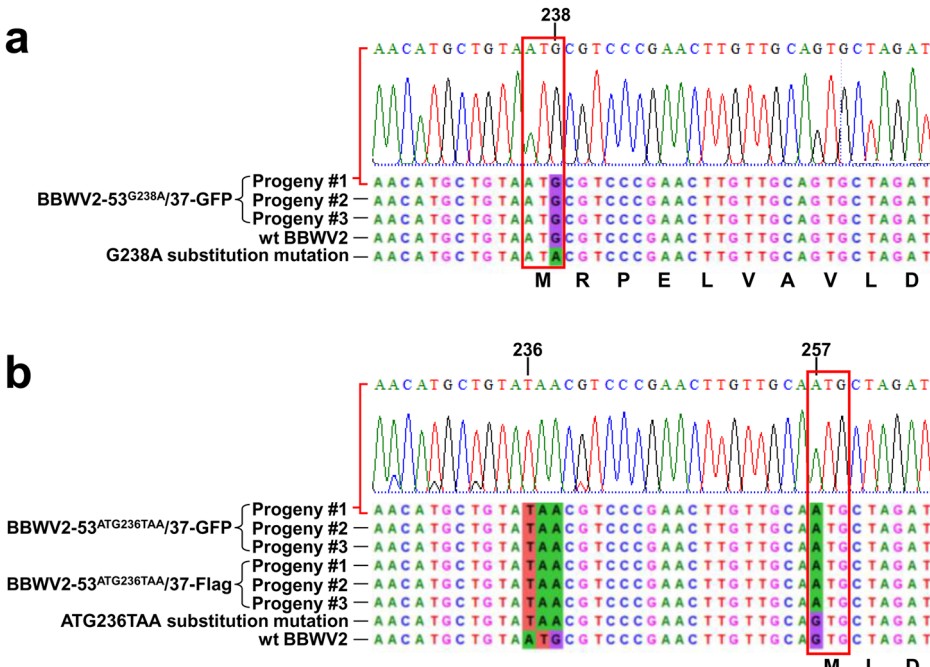

As an alternative approach to determine the requirement of VP53 in BBWV2 infection, we engineered two additional BBWV2 RNA2 mutant constructs, named pBBWV2-R2-Δ53N1-GFP and pBBWV2-R2-Δ53N2-GFP (Fig. 7a). pBBWV2-R2-Δ53N1-GFP involved the deletion of a significant genomic region corresponding to the N-terminus of VP53, resulting in the expression of a 17 amino acid peptide instead of the full-length VP53 (Fig. 7a). pBBWV2-R2-Δ53N2-GFP contained the deletion of the entire N-terminal VP53 sequence, which does not overlap with VP37 (Fig. 7a). To evaluate the infectivity of these mutant viruses, *N. benthamiana* plants were agroinfiltrated with pBBWV2-RP1-R1 and either pBBWV2-R2-Δ53N1-GFP or pBBWV2-R2-Δ53N2-GFP (these combinations were designated as BBWV2-Δ53N1-GFP and BBWV2-Δ53N2-GFP, respectively). The infectivity of the mutant viruses was assessed by examining the inoculated plants using a FOBI fluorescence imaging system at 8 dpi. While all the plants inoculated with BBWV2-53/37-GFP showed systemic infection, none of the plants inoculated with BBWV2-Δ53N1-GFP and BBWV2-Δ53N2-GFP were infected (Fig. 7b). Virus infection of the inoculated plants was confirmed through RT-PCR detection. The replication competence of the mutant viruses was assessed through the detection of replication-specific (-)-strand RNAs. Total RNA was isolated from the leaves infiltrated with BBWV2-53/37-GFP, BBWV2-Δ53N1-GFP, or BBWV2-Δ53N2-GFP at 2 dpi and analyzed by strand-specific RT-qPCR. Both mutants accumulated significantly reduced levels of (-)-strand RNA1 compared to BBWV2-53/37-GFP (Fig. 7c). More interestingly, the accumulation of (-)-strand RNA2 was dramatically diminished in both mutants (Fig. 7d). Consistently, Western blot analysis of the leaf samples infiltrated with BBWV2-Δ53N1-GFP and BBWV2-Δ53N2-GFP using anti-GFP antibodies revealed no detectable accumulation of VP37-GFP (Fig. 7e). Collectively, our results suggest the potential roles of VP53 in the replication of both RNA1 and RNA2 as well as viral systemic infection.

**The C-terminus of VP53, which overlaps with VP37, is dispensable for its function in facilitating the systemic infection of BBWV2**

To further explore the minimal region required for the function of VP53 in facilitating BBWV2 systemic infection, we first engineered a hybrid construct between pBBWV2-R2-Δ53N1-GFP and pBBWV2-R2-OE. Consequentially, this hybrid construct, named pBBWV2-R2-Δ53N1-OE, contained both a gene insertion cassette between the VP37 and LCP cistrons

and a deletion identical with that introduced in pBBWV2-R2-Δ53N1-GFP (Fig. 8a). Using pBBWV2-R2-Δ53N1-OE, we created a series of BBWV2 recombinant RNA2 constructs capable of expressing either the full-length VP53 or its C-terminal deletions from the gene insertion cassette (Fig. 8a). The recombinant RNA2 constructs were agroinfiltrated along with pBBWV2-RP1-R1 into *N. benthamiana* plants. Virus infection of the inoculated plants was determined by observing symptom development on the systemic leaves (Supplementary Fig. S5) and confirmed by RT-PCR detection. All the plants inoculated with BBWV2-Δ53N1-VP53, BBWV2-Δ53N1-VP53ΔC1, BBWV2-Δ53N1-VP53ΔC2, or BBWV2-Δ53N1-VP53ΔC3 displayed systemic infection (Supplementary Fig. S5). In particular, BBWV2-Δ53N1-VP53, BBWV2-Δ53N1-VP53ΔC1, and BBWV2-Δ53N1-VP53ΔC2 induced more severe symptoms (i.e., leaf size reduction, stunting, and vein yellowing) than wt BBWV2 and BBWV2-Δ53N1-VP53ΔC3 in *N. benthamiana* plants (Supplementary Fig. S5). However, none of the plants inoculated with BBWV2-Δ53N1-VP53ΔC4 were infected (Supplementary Fig. S5).

We also examined the accumulation levels of the BBWV2 recombinant viruses in systemically infected leaves. Total RNA was isolated from the upper symptomatic leaves of plants infected with wt BBWV2, BBWV2-Δ53N1-VP53, BBWV2-Δ53N1-VP53ΔC1, BBWV2-Δ53N1-VP53ΔC2, or BBWV2-Δ53N1-VP53ΔC3 at 10 dpi and analyzed by RT-qPCR. The accumulation levels of both RNA1 and RNA2 of BBWV2-Δ53N1-VP53, BBWV2-Δ53N1-VP53ΔC1, and BBWV2-Δ53N1-VP53ΔC2 were significantly higher compared to those of wt BBWV2 and BBWV2-Δ53N1-VP53ΔC3 (Fig. 8b, c). However, no significant difference in viral RNA accumulation was observed among BBWV2-Δ53N1-VP53, BBWV2-Δ53N1-VP53ΔC1, and BBWV2-Δ53N1-VP53ΔC2 (Fig. 8b, c). In addition, although the VP53 deletion mutant comprising amino acids 1 to 84 could support systemic infection of the virus (Supplementary Fig. S5), its activity in facilitating viral RNA accumulation was significantly diminished (Fig. 8b, c). The elevated accumulation of viral RNAs in BBWV2-Δ53N1-VP53 could be associated with the enhanced expression level of VP53, because VP53 expressed from the gene insertion cassette would be produced at an equivalent molar concentration as VP37 via proteolytic cleavage of the polyprotein. To examine this possibility, we generated an additional BBWV2 RNA2 hybrid construct containing a Flag tag at the C-terminus of VP37, named pBBWV2-R2-Δ53N1-F-OE (Fig. 9a). Using pBBWV2-R2-Δ53N1-F-OE, we engineered a BBWV2

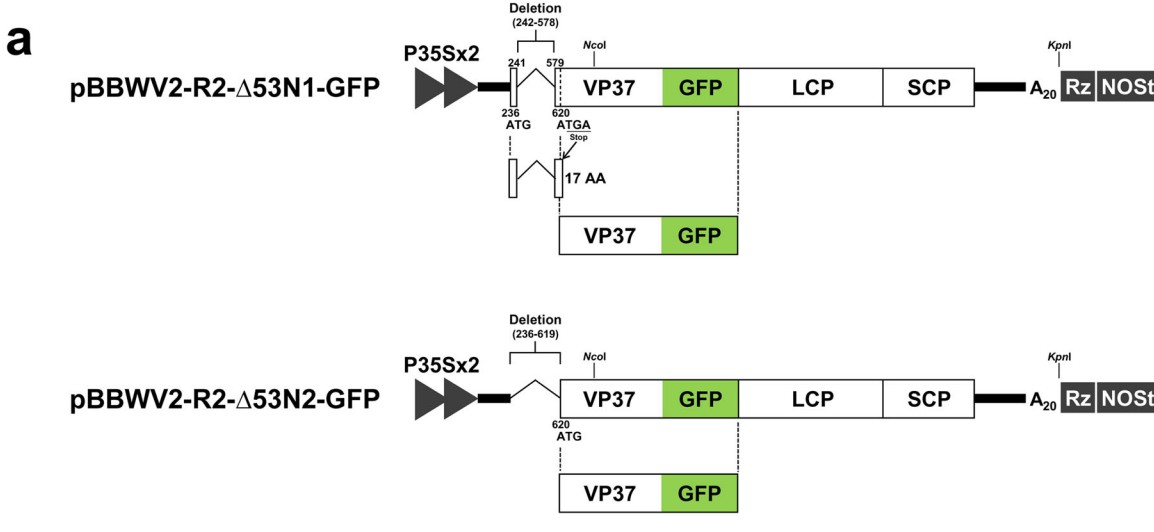

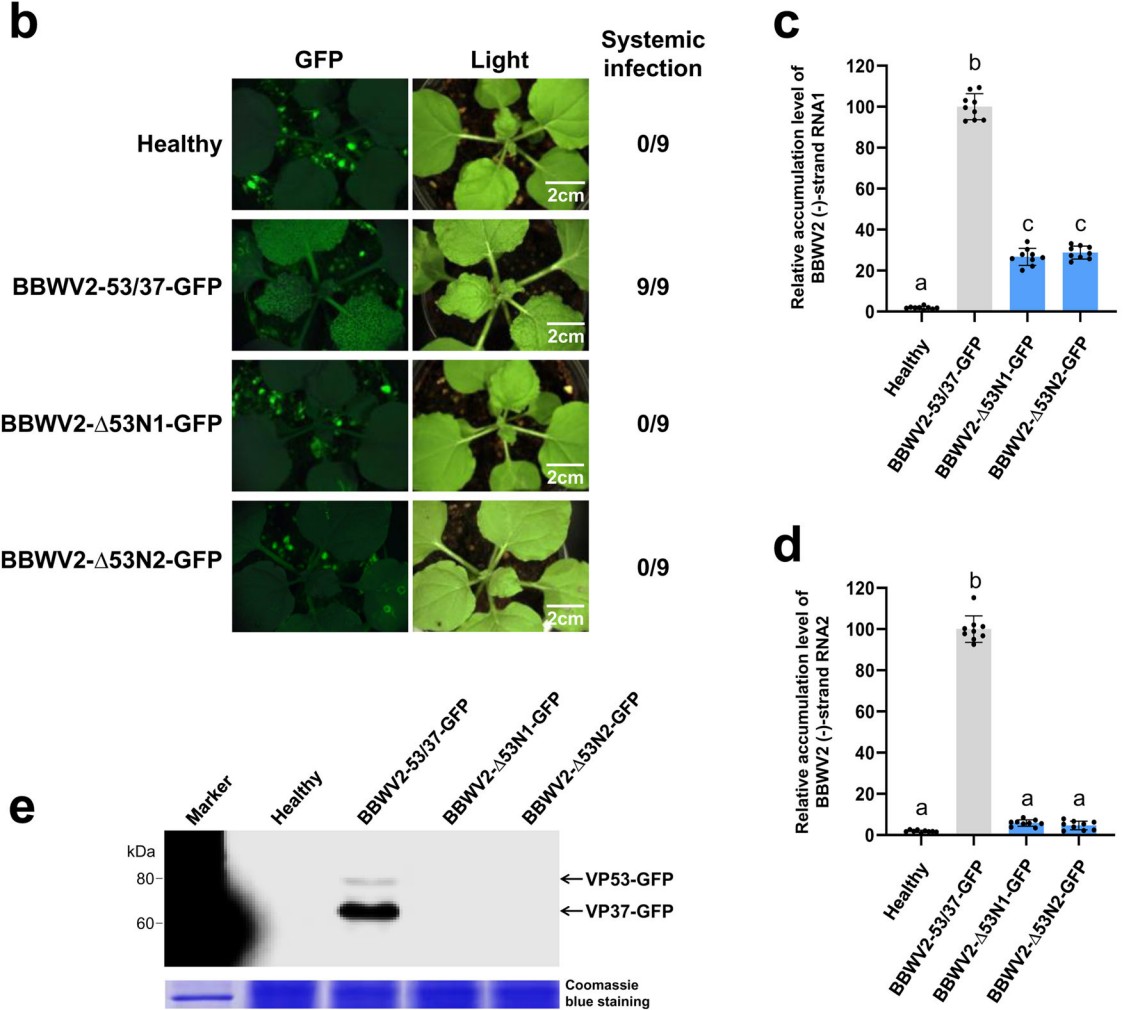

recombinant RNA2 construct denoted as pBBWV2-R2-Δ53N1-F-VP53:Flag, enabling the expression of VP53 with a Flag tag (Fig. 9a). *N. benthamiana* plants inoculated with BBWV2-Δ53N1-F-VP53:Flag exhibited systemic infection (Fig. 9b). As expected, Western blot analysis demonstrated a substantial elevation in the accumulation of VP53 when

expressed from the gene insertion cassette (i.e., BBWV2-Δ53N1-F-VP53:Flag) (Fig. 9c). Collectively, our results revealed that the C-terminal region of VP53, which overlaps with VP37, could be deleted without impairing its ability in facilitating systemic infection and viral RNA accumulation of BBWV2.

**Fig. 7 | VP53 is essential for systemic infection of BBWV2. a** Schematic representation of cDNA constructs of BBWV2 recombinant RNA2. pBBWV2-R2-Δ53N1-GFP involved the deletion of a significant genomic region corresponding to the N-terminus of VP53 (nucleotide positions 242-578), resulting in the expression of a 17 amino acid peptide instead of the full-length VP53. pBBWV2-R2-Δ53N2-GFP contained the deletion of the entire N-terminal VP53 sequence, which does not overlap with VP37. **b** Virulence of BBWV2-Δ53N1-GFP (pBBWV2-RP1-R1 + pBBWV2-R2-Δ53N1-GFP) and BBWV2-Δ53N2-GFP (pBBWV2-RP1-R1 + pBBWV2-R2-Δ53N2-GFP) in *N. benthamiana*. *N. benthamiana* plants inoculated with BBWV2-53/37-GFP, BBWV2-Δ53N1-GFP, or BBWV2-Δ53N2-GFP were observed using a FOBI fluorescence imaging system at 8 dpi. Data shown are representatives of three independent experiments, with each experiment involving the inoculation of three plants per viral construct. In total, nine plants were tested for the systemic infectivity of each virus (number of plants infected/number of plants inoculated). Virus infection of the inoculated plants was verified by RT-PCR

detection. Relative accumulation levels of (-)-strand RNAs of BBWV2 mutant viruses. Total RNA was isolated from *N. benthamiana* leaves inoculated with either BBWV2-53/37-GFP, BBWV2-Δ53N1-GFP, or BBWV2-Δ53N2-GFP at 2 dpi and subjected to strand-specific RT-qPCR to analyze the relative accumulation levels of (-)-strand RNA1 (**c**) and RNA2 (**d**). The mean ± SD of three replications is shown, and each column represents one group with nine plants. Significant differences indicated by different letters ($P < 0.05$) were analyzed using one-way ANOVA with Tukey's HSD test. **e** Western blot analysis of the expression of VP53-GFP and VP37-GFP. *N. benthamiana* plants were inoculated with either BBWV2-53/37-GFP, BBWV2-Δ53N1-GFP, or BBWV2-Δ53N2-GFP. Total protein was extracted from the inoculated leaves and subjected to immunoblot analysis using anti-GFP antibodies. Protein size markers are indicated on the left side of the blot. A Coomassie blue-stained gel is shown below the blot as a loading control. The blot image has been cropped for clarity, and the original blot image is presented in Supplementary Fig. S6.

## Discussion

The translation of viral proteins is a multifaceted process that viruses have evolved to overcome the limitations imposed by host translation systems and ensure efficient expression of their genetic information[1]. Most viruses possess compact genomes with limited coding capacity, necessitating the expression of multiple proteins from a single RNA molecule[2]. Overcoming this limitation is crucial for successful viral replication and propagation. One mechanism employed by viruses to express multiple proteins is ribosomal leaky scanning[1,5,29,30]. This process allows translation initiation to occur at non-canonical start codons, bypassing the conventional requirement for translation initiation at the first AUG codon. Thus, by utilizing ribosomal leaky scanning, viruses can translate downstream ORFs and effectively facilitate the expression of additional viral proteins.

The phenomenon of dual initiation of translation has been extensively studied in various viruses, including comoviruses (i.e., CPMV and BPMV) in the family *Secoviridae*[14,15,31]. Comoviruses exhibit high similarities with fabaviruses in terms of their genome size/structure, capsid composition, and virion morphology[32]. However, the limited amino acid sequence similarity (typically less than 30%) between comoviruses and fabaviruses justifies their classification into distinct genera[33]. Comoviruses are seed-borne and primarily transmitted by beetles in the family *Chrysomelidae*, whereas fabaviruses are transmitted by aphids and do not exhibit seed transmission[33]. In addition, while comoviruses usually have narrow host ranges, including several legume species, fabaviruses exhibit wide host ranges among dicotyledonous and some families of monocotyledonous plants[34]. In vitro and in vivo analyses demonstrated that translation initiation from both the first and the second AUG codons of CPMV RNA2 produces two largely overlapping polyproteins, with molecular weights of ~105 kDa and 95 kDa[15,31]. A previous study revealed that scanning ribosomes could bypass the AUG codon at nucleotide position 161 for the 105 kDa protein and initiate translation for the 95 kDa protein at the downstream AUG codon at nucleotide position 512 of CPMV RNA2[31]. Furthermore, it was shown that the 351 nucleotide sequence located between the two initiation codons possessed the capacity to direct ribosomes to initiate translation at a downstream start codon[31]. Based on these findings, it has been suggested that CPMV utilizes both leaky scanning and internal ribosomal entry mechanisms for the translation of RNA2. However, it has remained uncertain whether similar mechanisms are employed by fabaviruses.

In this study, we utilized infectious cDNA constructs of wt BBWV2 and its derivatives to examine the expression of VP53 during BBWV2 infection and to assess its functional role in BBWV2 infection. Genetic manipulation of infectious cDNA constructs of RNA viruses using the 35 S promoter and binary vectors has been widely employed to study the molecular biology of plant RNA viruses[35–38]. In this methodology, the 35 S promoter serves to generate initial viral RNA transcripts in inoculated plant leaves. The initial viral RNA transcripts are then translated and utilized as templates to initiate virus replication. Subsequently, the virus replicates under natural virus infection conditions, particularly in upper systemically infected leaves. Based on this approach, we demonstrated that BBWV2

RNA2 is translated to express two largely overlapping mature proteins, VP53 and VP37, during virus replication (Fig. 3b). Interestingly, the accumulation level of VP53 was found to be significantly lower than that of VP37 (Fig. 3b). In contrast, when VP53 was expressed from the gene insertion cassette (i.e., BBWV2-Δ53N1-F-VP53:Flag), the accumulation level of VP53 was observed to be nearly equivalent to that of VP37 (Fig. 9c). These findings suggest the possibility of a difference in translation efficiency between VP53 and VP37. The translation of varying quantities of viral proteins from a single viral mRNA is a typical phenomenon attributed to ribosomal leaky scanning because the efficiency of translation initiation at each AUG is influenced by the strength of the surrounding sequence context[1,5,29,30]. In higher plants, the optimal context includes most importantly an A at -3 and a G at +4 [cf. the consensus AUG context in higher plants: aA(A/C)aAUGGC; AUG itself corresponds to nucleotides +1 to +3][25,26]. If the context of the first AUG on the viral mRNAs is suboptimal, leaky scanning may occur[39]. Sequence analysis revealed that the start codon of both VP53 and VP37 were in suboptimal contexts; however, the context of the VP53 start codon having a G at -3 and a C at +4 is likely weaker than that of the VP37 start codon having an A at -3 and an A at +4 (Fig. 3c and d)[25,26]. Furthermore, as is well known in the ribosome shunting mechanism, which is a common feature among members of the family *Caulimoviridae*, ribosomes can bypass stem-loops containing AUG codons and reinitiate scanning at the 3' end of the stem-loops[39,40]. Interestingly, RNA structure analysis predicted that the VP53 start codon is located within a strong stem-loop RNA structure (Fig. 3e). In contrast, the VP37 start codon was predicted to be positioned between weak stem-loop RNA structures (Fig. 3f). The efficiency of translation initiation at a potential initiation codon can be enhanced when a stem-loop RNA structure is present immediately downstream of the initiation codon, thereby causing ribosomes to stall on the initiation codon[27,41,42]. These findings suggest that ribosomal leaky scanning, resulting from the low translation efficiency at the VP53 start codon, could facilitate the initiation of translation at the next downstream start codon for VP37. This hypothesis is supported by the observation that the accumulation level of VP53 was considerably lower than that of VP37 (Fig. 3b). We are currently conducting a separate study to further characterize the functional significance of the stem-loop RNA structures, which were predicted near the AUG start codons for VP53 and VP37, in regulating the translation efficiency of VP53 and VP37.

The BBWV2 recombinant viruses, namely BBWV2-Δ53N1-VP53, BBWV2-Δ53N1-VP53(1-298), and BBWV2-Δ53N1-VP53(1-128), remained fully infectious despite lacking the genomic region between the initiation codons for VP53 and VP37, which corresponds to the N-terminus of VP53 (Fig. 8 and Supplementary Fig. S5). These results suggest that the nucleotide sequences (at positions 242-578) between the two initiation codons are dispensable for VP37 translation. However, we cannot exclude the possibility that deleting the N-terminal region may affect the translation efficiency of VP37 by altering its 5'-untranslated region. In addition, we showed that an A-to-G substitution at nucleotide position 620 of BBWV2 RNA2 led to the disruption of VP37 translation but resulted in the

**Fig. 8 | The C-terminus of VP53, which overlaps with VP37, is not essential for its unique functionality. a** Schematic representation of cDNA clones of BBWV2 recombinant RNA2 constructs. pBBWV2-R2-Δ53N1-OE contains both a gene insertion cassette between the VP37 and LCP cistrons and the identical deletion in the VP53 N-terminus as introduced in pBBWV2-R2-Δ53N1-GFP. The gene insertion cassette includes an additional peptide cleavage site and two unique cloning sites (*Bgl*II and *Avr*II) as indicated. The amino acid sequences of the peptide cleavage sites recognized by Pro are underlined, and arrows indicate the position of the cleaved peptide bond. pBBWV2-R2-Δ53N1-VP53 allows for the expression of the full-length VP53 from the gene insertion cassette. pBBWV2-R2-Δ53N1-VP53ΔC1 allows for the expression of a C-terminal deletion mutant of VP53, encompassing amino acids 1 to 298, from the gene insertion cassette. pBBWV2-R2-Δ53N1-VP53ΔC2 allows for the expression of a C-terminal deletion mutant of VP53, encompassing amino acids 1 to 128, from the gene insertion cassette. pBBWV2-R2-Δ53N1-VP53ΔC3 allows for the expression of a C-terminal deletion mutant of VP53, encompassing amino acids 1 to 84, from the gene insertion cassette. pBBWV2-R2-Δ53N1-VP53ΔC4 allows for the expression of a C-terminal deletion mutant of VP53, encompassing amino acids 1 to 64, from the gene insertion cassette. Accumulation levels of BBWV2 recombinant viruses in systemically infected leaves. Total RNA was isolated from the upper symptomatic leaves of plants infected with wt BBWV2, BBWV2-Δ53N1-VP53, BBWV2-Δ53N1-VP53ΔC1, BBWV2-Δ53N1-VP53ΔC2, or BBWV2-Δ53N1-VP53ΔC3 at 10 dpi and subjected to RT-qPCR to analyze the relative accumulation levels of (+)-strand RNA1 (**b**) and RNA2 (**c**). The mean ± SD of three replications is shown, and each column represents one group with nine plants. Significant differences indicated by different letters ($P < 0.05$) were analyzed using one-way ANOVA with Tukey's HSD test.

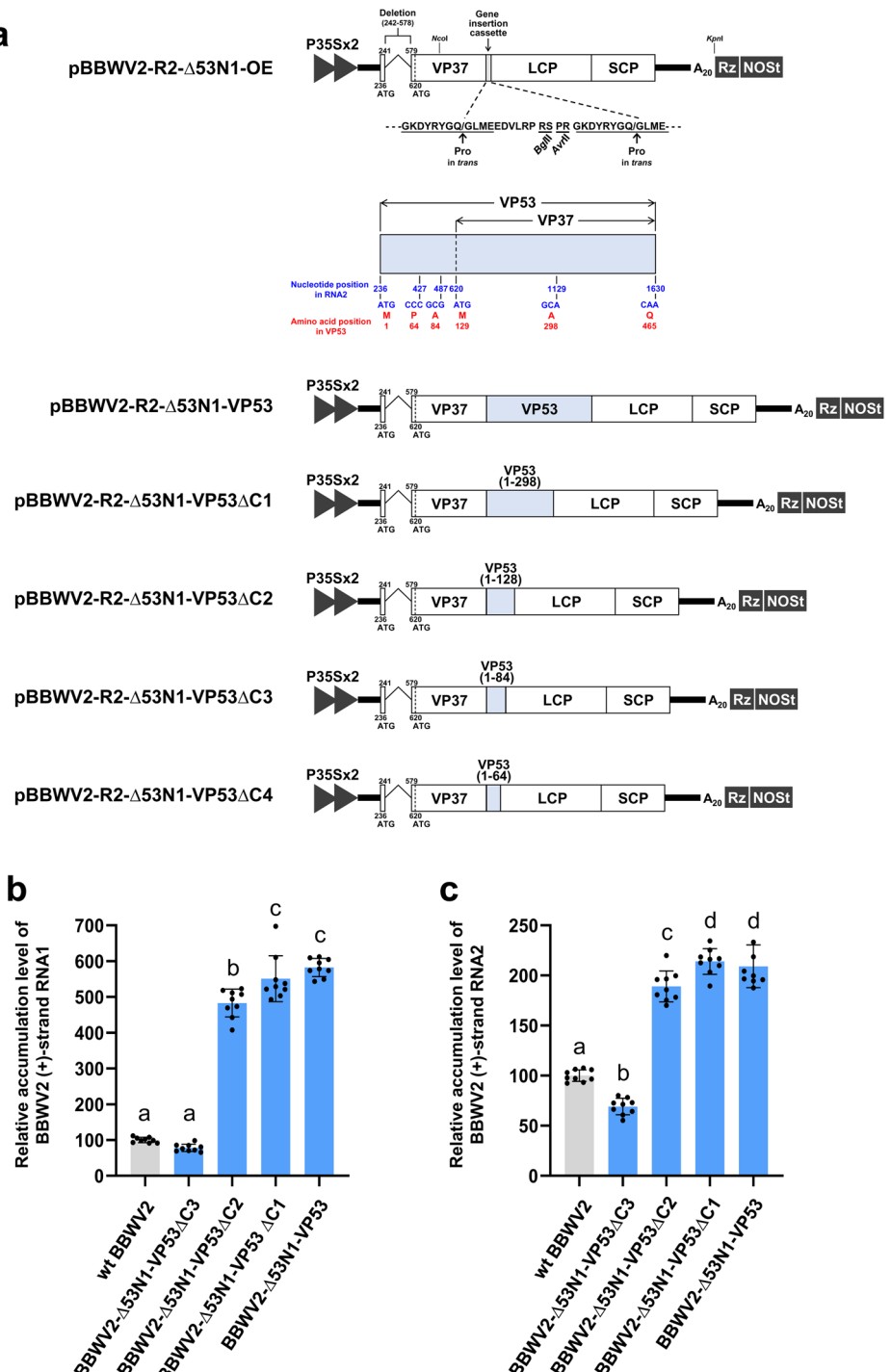

translation of a smaller 33 kDa protein, likely initiated at the next downstream ATG (at nucleotide positions 749-751) (Fig. 5c). This observation provides additional support for the phenomenon of ribosomal leaky scanning during the translation of BBWV2 RNA2. Collectively, our findings imply that BBWV2 is likely to lack the internal ribosomal entry capacity within the genome region between the AUG codons for VP53 and VP37, indicating that ribosomal leaky scanning serves as the primary mechanism in VP37 translation.

Plant virus MPs play a pivotal role in enabling the cell-to-cell and systemic movement of the viruses[43]. While the MPs of various virus species are known to predominantly localize to the cell periphery and PD, the subcellular distribution of MPs can vary among different virus species[43,44]. Consequently, investigating the subcellular distribution of MPs is important

to understand the mechanisms underlying the intracellular and intercellular virus movement. As expected, our confocal microscopic observation revealed that VP37 predominantly localized to the PD both in BBWV2-infected cells and when ectopically expressed via agroinfiltration (Figs. 1d and 2). This localization pattern indicates that the majority of VP37 proteins are likely associated with the formation of tubule structures within the PD. Furthermore, it appears that the VP37-GFP fusion protein retains its competence for tubule formation because BBWV2-53/37-GFP, which encodes VP37-GFP, exhibited successful systemic infection (Fig. 1c). In contrast, our observations revealed that VP53, despite being an N-terminally extended form of VP37, formed large cytoplasmic inclusions (Fig. 2). It is likely that the inclusion formation of VP53 was an artifact resulting from its ectopic overexpression via agroinfiltration because the

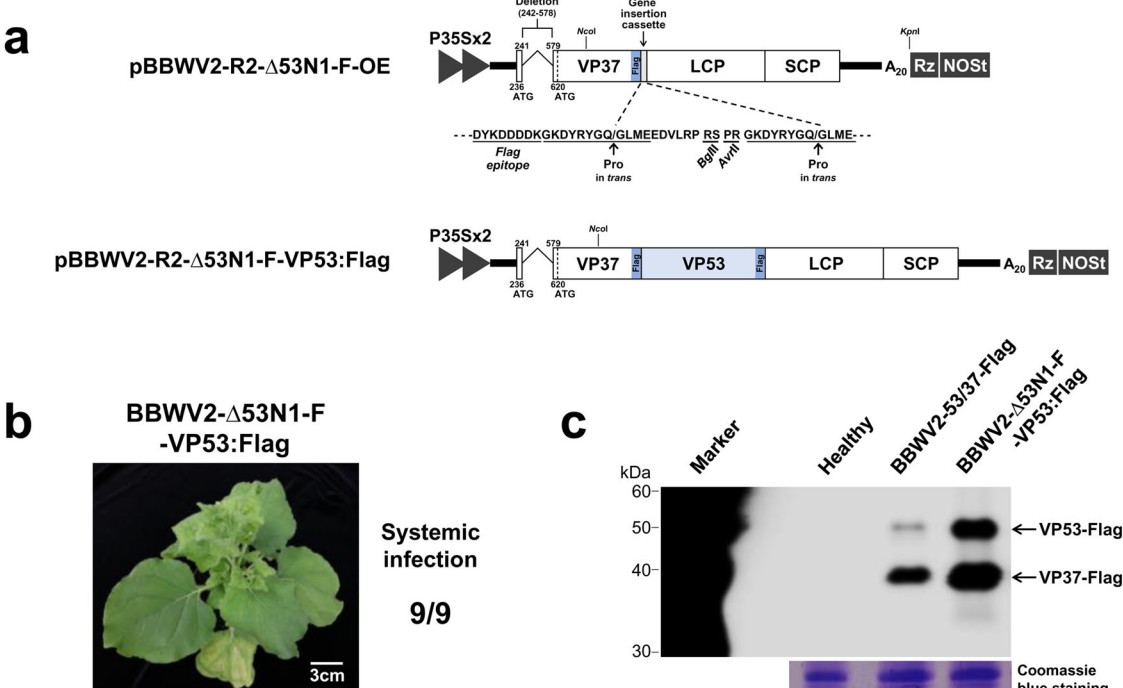

**Fig. 9 | Detection of the accumulation of VP53 when expressed from the gene insertion cassette of the BBWV2 recombinant RNA2. a** Schematic representation of cDNA constructs of BBWV2 recombinant RNA2 constructs. pBBWV2-R2-Δ53N1-F-OE contains the identical deletion in the VP53 N-terminus as introduced in pBBWV2-R2-Δ53N1-OE, the VP37 cistron fused in-frame with a Flag epitope, thereby expressing VP37-Flag, and a gene insertion cassette between the VP37 and LCP cistrons. pBBWV2-R2-Δ53N1-F-VP53:Flag allows for the expression of the full-length VP53 tagged with a Flag epitope from the gene insertion cassette. **b** Virulence of BBWV2-Δ53N1-F-VP53:Flag in *N. benthamiana*. *N. benthamiana* plants were inoculated with BBWV2-Δ53N1-F-VP53:Flag. Virus infection of the inoculated plants was determined by observing symptom development on the systemic leaves and confirmed by RT-PCR detection at 10 dpi. Data shown are representative of three independent experiments, with each experiment involving the inoculation of three plants per viral construct. In total, nine plants were tested for the systemic infectivity of each virus (number of plants infected/number of plants inoculated). **c** Western blot analysis of the expression of VP53 and VP37. *N. benthamiana* plants were inoculated with BBWV2-53/37-Flag or BBWV2-Δ53N1-F-VP53:Flag. Total protein was extracted from the upper symptomatic leaves of the inoculated plants and subjected to immunoblot analysis using anti-Flag antibodies. Protein size markers are indicated on the left side of the blot. A Coomassie blue-stained gel is shown below the blot as a loading control. The blot image has been cropped for clarity, and the original blot image is presented in Supplementary Fig. S6.

accumulation level of VP53 during virus replication was found to be extremely low (Fig. 3b). The discrepancy in the subcellular localization and accumulation levels between VP53 and VP37 suggests that VP53 may possess distinct structural functional properties compared to VP37 within the context of BBWV2 infection. Meanwhile, tagging with GFP at the C-terminus of VP53/VP37, but not with Flag, resulted in the enhancement of viral symptom severity in *N. benthamiana* plants (Figs. 1c and 4b). However, neither GFP- nor Flag-tagging affected the accumulation levels of viral RNAs (Supplementary Fig. S1). A previous study showed that the C-terminal motif of VP53/VP37 is responsible for the alteration of symptom severity in BBWV2[45]. A single amino acid substitution at the C-terminus of VP53/VP37 altered the pathogenicity of BBWV2-RP1 (a mild strain) to induce severe symptoms in *N. benthamiana* and pepper plants[45]. As the size of GFP (243 amino acids) is considerably larger than that of Flag (8 amino acids), GFP-tagging at the C-terminus of VP53/VP37 may affect the C-terminal protein structures of VP53/VP37 and/or the accessibility of other interacting partner proteins, potentially resulting in the enhancement of the symptom severity of BBWV2. Further studies involving a comparative analysis of the protein structures of VP53 and VP37 are needed to elucidate their specific roles and functions in the viral replication cycle and pathogenicity.

In this study, we have successfully demonstrated that VP53 is crucial for facilitating systemic infection of BBWV2. Interestingly, our findings also revealed that the function of VP53 is distinct from that of VP37, as most of the VP37 region could be deleted from VP53 without compromising virus replication and infectivity (Fig. 8 and Supplementary Fig. S5). The requirement of VP53 in virus replication and systemic infection was further highlighted by the restoration of the G-to-A substitution at position 238 and the occurrence of the compensatory G-to-A mutation at position 257 generating a new start codon downstream of the original one (Fig. 6). In addition, the new start codon that begins at position 257 would produce an N-terminally 7-amino-acid-truncated form of VP53 (Fig. 6b). Therefore, it is possible that the translation efficiency of the sequence context of the new start codon may differ from that of the original start codon for VP53. Indeed, the accumulation level of VP53 in the leaves infected with the progenies of BBWV2-53$^{ATG236TAA}$/37-Flag appeared to be slightly higher than that in the leaves infected with BBWV2-53/37-Flag (Fig. 5d). Meanwhile, the A-to-G substitution at nucleotide position 620 was not restored despite the mutant virus BBWV2-R2-53/37$^{A620G}$-Flag was actively replicated in the inoculated leaves (Fig. 5c). In addition, the original mutations engineered into BBWV2-53$^{ATG236TAA}$/37-GFP and BBWV2-53$^{ATG236TAA}$/37-Flag were present in all progeny viruses obtained from systemically infected leaves (Fig. 6b). Interestingly, a previous study demonstrated that BPMV P58, a counterpart of BBWV2 VP53, is required *in cis* for RNA2 replication[14]. In this study, we observed that the BBWV2 RNA2 mutants, unable to express VP53, exhibited significant reductions in viral replication (Fig. 7c, d). Conversely, the increased expression of VP53 from the gene insertion cassette resulted in an elevated accumulation of both RNA1 and RNA2, consequently exacerbating symptom severity (Figs. 8, 9c and Supplementary Fig. S5). Therefore, our results suggest that BBWV2 VP53 may function in facilitating BBWV2 RNA replication.

Our study provides important insights into the expression and function of VP53 and VP37 during BBWV2 infection. The findings presented herein emphasize the distinct subcellular localization of these proteins and

their differential contribution to virus infectivity. Our mutational analyses revealed that VP53 and VP37 are translated from BBWV2 RNA2 via ribosomal leaky scanning. In particular, we demonstrated the requirement of VP53 in the replication and systemic infection of BBWV2. Understanding the roles of individual viral proteins is essential for unraveling the molecular mechanisms underlying viral replication and pathogenesis, ultimately facilitating the development of strategies for virus control and crop protection.

## Methods

### Generation of the recombinant and mutant constructs of BBWV2 RNA2

A modified cDNA clone of BBWV2-RP1 RNA2 pBBWV2-R2-OE[23]; was engineered to generate various BBWV2 RNA2 recombinant and mutant constructs. A 1361-bp DNA fragment comprising the C-terminal half of VP53 (from the NcoI site) and the green fluorescent proteon (GFP) coding sequence was synthesized (Macrogen, Korea) and inserted into the pBBWV2-R2-OE vector, which was digested with NcoI and AvrII (Fig. 1b). The resulting construct was named pBBWV2-R2-53/37-GFP (Fig. 1b). Similarly, a 659-bp DNA fragment comprising the C-terminal half of VP53 (from the NcoI site) and a Flag epitope sequence (GACTACAAGGACGACGATGACAAG) was synthesized (Macrogen, Korea) and inserted into pBBWV2-R2-OE, which was digested with NcoI and AvrII (Fig. 3a). The resulting construct was named pBBWV2-R2-53/37-Flag (Fig. 3a). To introduce specific nucleotide substitution mutations into the initiation codon for VP53 or VP37, we modified pBBWV2-R2-53/37-GFP and pBBWV2-R2-53/37-Flag based on a fusion-PCR-based site-directed mutagenesis strategy using overlapping primers (Supplementary Table S1)[46]. To construct a cDNA clone of BBWV2 recombinant RNA2, which includes the deletion of a significant genomic region corresponding to the N-terminus of VP53 (nucleotide positions 242-578), we employed a fusion-PCR-based cloning strategy using overlapping primers (Supplementary Table S1). Briefly, the 5′ genomic region of BBWV2 RNA2 from nucleotide 1 to 241 was amplified using pBBWV2-R2-53/73-GFP as a template and the corresponding primers (Supplementary Table S1; BBWV2R2-5E-Fw + Δ53N1-Rv). The genomic region of BBWV2 RNA2 from nucleotide 579 to the 3′ end was amplified using pBBWV2-R2-53/73-GFP as a template and the corresponding primers (Supplementary Table S1; Δ53N1-Fw + pBBWV2-R2-3E-KpnI-Rv). These PCR products were then used as templates in a subsequent fusion-PCR, using the corresponding primers (Supplementary Table S1; BBWV2R2-5E-Fw + pBBWV2-R2-3E-KpnI-Rv). The resulting PCR product was cloned into between the StuI and KpnI sites in the pCass-Rz vector, as described in our previous study [23,47]. The resulting construct was named pBBWV2-R2-Δ53N1-GFP (Fig. 7a). An additional cDNA clone of BBWV2 recombinant RNA2, which includes the deletion of the entire N-terminal VP53 sequence (nucleotide positions 236-619) that does not overlap with VP37, was generated by a similar fusion-PCR-based strategy using overlapping primers (Supplementary Table S1). The resulting construct was named pBBWV2-R2-Δ53N2-GFP (Fig. 7a). The construction of pBBWV2-R2-Δ53N1-OE, which contains a gene insertion cassette between the VP37 and LCP cistrons and the identical deletion in the VP53 N-terminus as introduced in pBBWV2-R2-Δ53N1-GFP, was performed as follows: the genomic region of BBWV2 RNA2 from nucleotide 1002 (the NcoI site) to the 3′ end was obtained by digesting pBBWV2-R2-OE with NcoI and KpnI[23]. The resulting fragment was cloned into the NcoI and KpnI sites in pBBWV2-R2-Δ53N1-GFP (Figs. 7a, 8a). The generation of BBWV2 recombinant RNA2 constructs capable of expressing either the full-length VP53 or its C-terminal deletions from the gene insertion cassette was performed as follows: the full-length VP53 and its C-terminal deletions were amplified using corresponding primer pairs (Supplementary Table S1). The resulting PCR products were digested with BglII and AvrII and inserted into pBBWV2-R2-Δ53N1-OE, which was digested with BglII and AvrII (Fig. 8a). To construct pBBWV2-R2-Δ53N1-F-OE, a 713-bp DNA fragment comprising the C-terminal half of VP53 (from the NcoI site), a Flag epitope sequence, and an additional Pro cleavage

sequence was synthesized (Macrogen, Korea) and inserted into the pBBWV2-R2-Δ53N1-OE, which was digested with NcoI and BglII (Fig. 9a). The generation of pBBWV2-R2-Δ53N1-F-VP53:Flag, capable of expressing the full-length VP53 tagged with a Flag epitope from the gene insertion cassette, was performed as follows: the full-length VP53 containing a Flag epitope sequence at the C-terminus was amplified using the corresponding primer (VP53-BglII-Fw + VP53-Flag-AvrII-Rv; Supplementary Table S1). The resulting PCR products were digested with BglII and AvrII and inserted into pBBWV2-R2-Δ53N1-F-OE, which was digested with BglII and AvrII (Fig. 9a). Plasmid DNAs of the generated constructs were transformed into the Agrobacterium strain EHA105.

### Generation of constructs for the ectopic expression of GFP-fused VP37 and VP53

The PZP-GFP vector, which enables in planta expression of green fluorescence protein (GFP) via Agrobacterium-mediated infiltration (agroinfiltration)[24]. The coding sequences of VP53 and VP37 were amplified by PCR using the appropriate primer sets (Supplementary Table S1) and inserted in-frame upstream of the GFP gene in the PZP-GFP vector utilizing StuI and SpeI sites. The resulting constructs were named PZP-VP53-GFP and PZP-VP37-GFP, respectively. Plasmid DNAs of the generated constructs were transformed into the Agrobacterium strain EHA105.

### Plant growth, viral sources, inoculation, and tissue staining

Nicotiana benthamiana plants were cultivated in an insect-free growth chamber under controlled conditions with a photoperiod of 16 h of light at 26 °C and 8 h of darkness at 24 °C. Full-length cDNA clones of BBWV2-RP1 and its derivative constructs generated in this study were used as viral sources and inoculated via agroinfiltration into leaves of two-week-old N. benthamiana plants[23,45,48]. For agroinfiltration, T-DNA-based binary vector constructs were transformed into Agrobacterium strain EHA105. Agrobacteria harboring each plasmid constructs were grown at 30 °C overnight in YEP medium containing kanamycin (100 μg/mL) and acetosyringon (20 μM). The Agrobacteria were harvested by centrifugation at 3000 rpm for 10 min, resuspended in the infiltration buffer (MS salts, 10 mM MES, pH 5.6, 200 μM acetosyringon) to 0.5 $OD_{600}$, and incubated at 30 °C for 4 h. Agrobacterium cultures were infiltrated onto the abaxial surface of leaves of N. benthamiana plants using a 1-ml syringe. In each inoculation experiment, we conducted three independent experiments, each involving a minimum of three plants per viral construct. To stain the PD, N. benthamiana leaves were infiltrated and incubated with a solution of 0.1 mg/ml aniline blue (Sigma-Aldrich, USA) for 30 min[49].

### Fluorescence imaging

The visualization of GFP fluorescence signals in living plants was performed using a FOBI fluorescence imaging system (NeoScience, Korea) equipped with a blue light source (excitation at 470 nm) and an emission filter (530 nm short-pass), which effectively eliminates autofluorescence signals from chlorophyll[22]. Cellular fluorescence signals in plant leaf tissues were observed using a Leica SP8 laser-scanning confocal microscope (Leica, Germany) equipped with specific laser/filter combinations for GFP (excitation at 488 nm, detection between 510 and 550 nm), aniline blue (excitation at 405 nm, detection between 460 and 500 nm), and chloroplasts (excitation at 568 nm, detection between 580 and 620 nm).

### Western blot analysis

Total protein was extracted from N. benthamiana leaves using TRIzol (Invitrogen, USA) following the manufacturer's protocol. Proteins were separated by 10% SDS-PAGE and subsequently transferred onto a polyvinylidene difluoride membrane. The membrane was then probed with either anti-Flag or anti-GFP antibodies (Invitrogen, USA). For visualization of the antigens, a secondary antibody conjugated to horseradish peroxidase (Sigma-Aldrich, USA) was used with the Amhersham ECL Western Blotting Detection System (GE Healthcare Life Sciences, USA).

## Liquid chromatography coupled with tandem mass spectrometry (LC-MS/MS) analysis

Total protein extracted from *N. benthamiana* leaves infected with BBWV2-53/37-Flag was separated by 10% SDS-PAGE. A gel fragment corresponding to the molecular weight of VP53-Flag (~53 kDa) was excised from the gel and subjected to in-gel digestion using trypsin followed by LC-MS/MS analysis. The entire LC-MS/MS analysis was performed at Yonsei Proteome Research Center (Seoul, South Korea). Briefly, LC was performed using an Easy n-LC 1000 system (Thermo Fisher Scientific, USA). Peptide separation was performed using a C18-nanobore column (150 mm × 0.1 mm, 3-μm pore size, Agilent, USA). LTQ-Orbitrap mass spectrometry (Thermo Fisher, USA) was used to identify and quantify peptides. Xcalibur (version 2.1, Thermo Fisher Scientific, USA) was used to generate peak lists. The peak lists were identified by searching the National Center for Biotechnology Information database using the MASCOT search engine (http://www.matrixscience.com, Matrix Science, USA).

## Sequence analyses

Sequence alignments of various virus isolates were generated using ClustalW implemented in MEGA X software[50] and analyzed using WebLogo to obtain consensus sequences surrounding translation initiation sites[51]. RNA secondary structures were predicted using RNAfold[52].

## RNA extraction, virus detection, sequencing, and quantification

RNA extraction was performed using the PureLink RNA Mini kit (Ambion, USA) following the manufacturer's protocol. To verify systemic infection of the inoculated plants with BBWV2, total RNA was extracted from the upper uninoculated leaves of the plants and analyzed by RT-PCR using a BBWV2-specific primer pair (5′-CAGAGAAGTGGTTGGTCCCGTG-3′ and 5′-ATGGGAGGCTAGTGACCTACG-3′)[11]. The 5′ genomic regions of progeny viruses derived from the nucleotide substitution mutant viruses were amplified by RT-PCR using an appropriate primer pair (5′-ACAAACAGCTTTCGTTCCGAAA-3′ and 5′-GTAGCTAGGAACGTTCTTGCT-3′). The resulting PCR products were subsequently subjected to Sanger sequencing. The sequence data were analyzed using ClustalW implemented in MEGA X software[50]. To quantify the accumulation levels of (+)-strand BBWV2 RNAs, RT followed by quantitative PCR (RT-qPCR) was performed using the 2X SYBR Green Real-Time PCR Smart mix (Solgent, Korea) and iCycler iQ5 detection system (Bio-Rad, USA) with the following specific primers: BBWV2-R1-Fw (5′-TCACAGGTTATGCCGCTTGT-3′) and BBWV2-R1-Rv (5′-TCACTCGTCCCAAGCTGTTC-3′) for BBWV2 (+)-strand RNA1 detection; BBWV2-R2-Fw (5′-CCAGAGAAGTGGTTGGTCCC-3′) and BBWV2-R2-Rv (5′-TCCAACAGGTAATGCCCACC-3′) for BBWV2 (+)-strand RNA2 detection[22]. Strand-specific RT-qPCR was performed to quantify the accumulation levels of (-)-strand BBWV2 RNAs. Briefly, cDNAs of (-)-strand RNA1 and RNA2 were synthesized using SuperScript IV (Invitrogen, USA) with tagged RT primer: tag-R1(-)-RT (5′-GCTGGAATTCGCGGTTAAATCACAGGTTATGCCGCTTGT-3′) for cDNAs synthesis of (-)-strand RNA1 and tag-R2(-)-RT (5′-GCTGGAATTCGCGGTTAAACCAGAGAAGTGGTTGGTCCC-3′) for cDNAs synthesis of (-)-strand RNA2. Each tagged RT primer contains a non-BBWV2 sequence (boldface) at the 5′ end. The resulting cDNA was purified using the GENECLEAN Turbo kit (MP Biomedicals, USA) to remove residual tagged RT primers and then subjected to qPCR. qPCR was performed using the Luna Universal qPCR Master Mix (NEB, USA) and iCycler iQ5 detection system with the following specific primers: tagPR (5′-GCTGGAATTCGCGGTTAAA-3′) and BBWV2-R1-Rv for BBWV2 (-)-strand RNA1 detection; tagPR and BBWV2-R2-Rv for BBWV2 (-)-strand RNA2 detection. The actin gene was analyzed as an internal reference control using Nb-actin-qRT-Fw (5′-CGAGGAGCATCCAGTCCTCT-3′) and Nb-actin-qRT-Rv (5′-GTGGCTGACACCATCACCAG-3′) to normalize the results of RT-qPCR and strand-specific RT-qPCR assays. Three biological and three technical replicates were analyzed per sample.

## Statistics and reproducibility

All experimental results in the manuscript are representatives of three independent experiments. Each experiment was performed using at least three plants per condition. Statistically significant differences among experimental groups were analyzed using one-way ANOVA with Tukey's HSD test ($P < 0.05$). The mean ± SD of three replications is shown, and each column represents one group with at least nine plants.

## Reporting summary

Further information on research design is available in the Nature Portfolio Reporting Summary linked to this article.

## Data availability

The original contributions presented in the study are included in the article/ Supplementary material. The source data behind the graphs can be found in Supplementary Data 1. Uncropped blots can be found in Supplementary Fig. S6. All other data are available from the corresponding author on reasonable request.

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

## Acknowledgements

This research was supported in part by grants from Agenda Program (PJ015308) funded by the Rural Development Administration of Korea, Basic Science Research Program (NRF-2022R1A2C1004728) funded by the National Research Foundation of Korea, and 2023 Research Grant Program (A100-20230040) funded by Seoul National University.

## Author contributions

J.K.S. designed the experiments and supervised the project; M.H.K., B.C., S.Y.J., J.S.C., S.K., Y.L., S.P. and S.J.K. performed the experiments; M.H.K., B.C., S.J.K., J.H.K. and J.K.S. analyzed the data; M.H.K., B.C., S.J.K. and J.K.S. wrote and revised the manuscript.

## Competing interests

The authors declare no competing interests.
