## [Peer review file · Communications Biology]

Referee expertise:

Referee #1: plant-virus interactions

Referee #2: plant-virus interactions

Referee #3: plant-virus interactions

Reviewers' comments:

Reviewer #1 (Remarks to the Author):

In this research, the authors attempted to elucidate the function of BBWV2 VP57 in virus infection, and made a claim on its indispensable role.

1. VP57, as well as VP37 and the large capsid protein, have been demonstrated to be RNA silencing suppressors (Kong et al., 2014 Virus Res, 192:62-73). This research does not reveal any new function of VP57. The molecular mechanisms of how VP57 promote virus pathogenicity is also largely missing.
2. The authors complicated their experimental design by carrying out a lot of mutation and epi-tag fusing assays. However, these assays did not help a lot in elucidating mechanisms. The claim that VP57 is indispensable for VP37 expression and function primarily came from the deletion of VP57 N terminal. Although the deletion does not disrupt VP37 ORF, this region may still affect VP37 expression as its 5' UTR. Thus, the observation that its deletion impaired VP37 expression cannot justify the claim that VP57 is required for VP37 expression and function.
3. Due to the extremely low expression levels of VP57 protein and its different subcellular localization from that of VP37, it is likely that VP57 is only a secondary backup of VP37 that plays its roles inside the plant cell instead of plasmodesmata. This research exaggerated the role and function of VP57.

Reviewer #2 (Remarks to the Author):

Authors study in detail the existence and possible function of the putative VP53 protein of the fabavirus Broad bean wilt virus 2 (BBW2). They engineer several recombinant and mutant constructs in order to decipher the involvement of this protein in the viral infection. They provide evidence of the possible mechanism of translation initiation and show that the protein is essential for the life cycle of BBW2. I think they present very interesting and powerful results to support their conclusions, like the ones obtained with the natural compensatory mutations found after removal of the start codon of VP53 or the experiments in which VP53 is expressed independently after VP37.

The part that I have found less clear is the one about the leaky expression of the proteins. And this is probably because of two things. The first one is that it is not clear to me what is the starting hypothesis of the authors in this regard. Do they thought BBWV2 would behave like their comovirus cousins, using a dual mechanism for the expression of the VP proteins or do they thought that leaky expression would be the most likely scenario for BBWV2? I think the explanation of the results vary depending if it is one or the other. The second one I think is because of the experiments performed *in silico* with the AUG surrounding regions. These results are presented in Figure 4C-F, but they are not mentioned in the discussion. Thus, I do not know if they actually match the leaky expression hypothesis or not. Besides, in the *in silico* analyses of the start initiation, authors state

that “nucleotide sequences around both start codons were highly conserved among BBWV2 isolates”. I can see this for VP37, but not in the case of VP53, at least not in the part upstream the AUG, where only 4 out of 10 nucleotides are conserved. Then authors say that the sequences around AUG are unfavorable for eukaryotic ribosome recognition, and that this is surprising. However, to my knowledge, authors do not provide any possible explanation for this in the results or in the discussion. Authors also perform secondary structure analysis in these regions and they found hairpin RNA structures they consider intriguing. Again, do these data fit the leaky expression hypothesis or not? This is not clearly explained either in my opinion.

Besides this, there are other minor comments or concerns that I address below:

- I think when authors state in lines 330-331 that “Our analysis clearly detected both VP53-Flag and VP37-Flag at the expected size” they are too categorical. They observe bands at the anticipated size of VP53-Flag and VP37-Flag. However, for example the band matching the size of VP53-Flag could be VP37-Flag with some posttranslational modifications, or it could be something else. Unlikely, but possible. Mass spectrometry could be a good complement to try to identify N-terminal peptides of VP53.
- Probably Figs. 1, 2 and 3 or at least Figure 1 and 2 could be merged together.
- In Fig. 2C, why there is no Aniline blue signal in the healthy plant?
- In Fig. 4B, Fig. 6C and D and Fig. 10E, I think the labels VP37 and VP53 should be VP37-FLAG and VP53-FLAG.
- In Fig. 7 it is not clear to me to which sequence corresponds the electropherogram that appears on top of the alignments.
- In Fig 9B I am not able to see any differences between infected and not-infected plants. Probably it could be a good option to move these images to supplementary and include Fig 10A and 10B in this Figure 9 instead.
- Line 107: VP53 is essential...
- Line 111: remove the “;” after pBBWB2-R2-OE
- Why a WB is not performed using anti-GFP, as in Fig 8E, to verify the presence of VP53 instead of making specific constructs with Flag? Wouldn't it be more informative to have the data of the presence of VP53 in samples already analyzed by confocal microscopy?
- In lines 340-343, the word “sequence” appears too many times.
- The fact that GFP exacerbated the symptoms and Flag did not is not discussed. Do the authors have any hypothesis to explain this?

Reviewer #3 (Remarks to the Author):

The study analyzed the protein VP53 potentially encoded by Broad bean wilt virus 2 (BBWV2) using genetic, virological, cell biology, and molecular biology approaches to investigate the functionality and expression of VP53. Overall, the research provides new data for a deeper understanding of BBWV2 pathogenesis. I have a few suggestions to further improve the quality of this manuscript:

1. To confirm that VP53 is indeed encoded and expressed by BBWV2, it is recommended to use a VP53-specific antibody (e.g., N-terminal specific antibody against VP53) for further validation of VP53 expression.
2. line 329, “*N. benthamiana*” should be italicized. Check the entire manuscript for similar errors.

3. line 353, I wonder whether the hairpin RNA structures affect the translation of both VP53 and VP37.
4. line 383, avoid stating "data not shown." Provide the relevant data to support the claims made in the paper.
5. line 426, since the mentioned mutations did not completely block VP53 expression, I suggested introducing more stop codons at the N-terminus of VP53 and examine their impact on BBWV2 infection.
6. line 471, I wonder whether the deletion of the N-terminus of VP53 (Fig. 8A) affects the RNA structure, thereby influencing BBWV2 infection.
Merge Fig. 1 and Fig. 2 for better presentation.
7. In Fig. 6D, explain why the expression of VP53 in BBWV2-53ATG236TAA/37-Flag is higher than in BBWV2-53/37-Flag, it seems that the mutation enhances VP53 expression.
8. For Fig. 7, consider using RNA-seq (deep sequencing) to further characterize the mutations of BBWV2-53 mutants during infection.
9. I wonder how about the mutation of residue 257 to a stop codon on BBWV2 infection.
10. Fig. 9B, whether these mutations affect the symptom severity of BBWV2? and molecular detection should be conducted to verify the symptom phenotype shown in Fig. 9B.

Reviewers' comments:

<Reviewer #1>

In this research, the authors attempted to elucidate the function of BBWV2 VP57 in virus infection, and made a claim on its indispensable role.

1. VP57, as well as VP37 and the large capsid protein, have been demonstrated to be RNA silencing suppressors (Kong et al., 2014. *Virus Res*, 192:62-73). This research does not reveal any new function of VP57. The molecular mechanisms of how VP57 promote virus pathogenicity is also largely missing.

→ Thank you for the comments. One of the key findings of our study is that VP53 is indeed expressed during BBWV2 infection. So far, fabavirus RNA2 is hypothesized to encode two largely overlapping ORFs using two alternative in-frame initiation codons, but this has not yet been demonstrated: the expression and function of VP37 have been demonstrated in previous studies (Xie et al., 2016, *Sci Rep* 6:21552; Liu et al., 2011, *Virus Res* 155:42-47); however, it remains unproven whether VP53 is truly expressed during viral replication and essential for successful infection of BBWV2. In this study, we demonstrated that VP53 was detectably expressed during BBWV2 infection (Figs. 3B, 5C, 5D, and S2). Furthermore, we showed that the removal of the N-terminal non-overlapping region of VP53 from RNA2 resulted in BBWV2 being unable to systemically infect plants (Fig. 7). We also showed that the C-terminal region of VP53, which overlaps with VP37, is dispensable for its ability in facilitating systemic infection and viral RNA accumulation of BBWV2 (Fig. 8 and Supplementary Fig. S5). We believe that our findings provide a novel insight on the genetic organization of BBWV2 and contribute to our understanding of the infection mechanisms employed by fabaviruses.

2. The authors complicated their experimental design by carrying out a lot of mutation and epi-tag fusing assays. However, these assays did not help a lot in elucidating mechanisms. The claim that VP57 is indispensable for VP37 expression and function primarily came from the deletion of VP57 N terminal. Although the deletion does not disrupt VP37 ORF, this region may still affect VP37 expression as its 5' UTR. Thus, the observation that its deletion impaired VP37 expression cannot justify the claim that VP57 is required for VP37 expression and function.

→ Thank you for the comments. However, there may be a misunderstanding. We did not claim that VP57 is indispensable for VP37 expression and function, but our results clearly showed that the C-terminus of VP53, which overlaps with VP37, is dispensable for its function in facilitating systemic infection of BBWV2, as shown by testing with BBWV2- Δ 53N1-VP53 Δ C2 (Fig. 8 and Supplementary Fig. S5). For the deletion of the genomic region corresponding to the N-terminus of VP53, we claimed that VP53 is required for BBWV2 RNA replication, particularly for RNA2, because the accumulation of (-)-strand RNA2 was almost completely diminished in both deletion mutants (BBWV2- Δ 53N1-GFP and BBWV2- Δ 53N2-GFP) (Fig. 7D). Therefore, because the replication of the deletion mutants was dramatically reduced, the expression of VP37 was not detectable in Fig. 8E.

3. Due to the extremely low expression levels of VP57 protein and its different subcellular localization from that of VP37, it is likely that VP57 is only a secondary backup of VP37 that plays its roles inside the plant cell instead of plasmodesmata. This research exaggerated the role and function of VP57.

→ VP37 is well characterized as the movement protein that forms tubules in the plasmodesmata (PD) to mediate the tubule-guided cell-to-cell movement of BBWV2 virions (Xie et al., 2016, *Sci Rep* 6:21552; Liu et al., 2011, *Virus Res* 155:42-47). As shown in Fig. 2, VP37 predominantly accumulated in the PD, whereas VP53 was observed in the cytoplasm but not in the PD. To function as a viral movement protein facilitating the tubule-guided cell-to-cell movement, a viral protein must localize to the PD. However, VP53 does not exhibit PD localization, indicating that it does not function as a movement protein like VP37. Furthermore, the C-terminus of VP53, which overlaps with VP37, is dispensable for its function in facilitating systemic infection of BBWV2 (Fig. 8 and Supplementary Fig. S5), suggesting that the function of VP53 is different from that of VP37. Our results showed that VP53 is a functional viral protein required for BBWV2 infectivity, although its accumulation level was significantly lower than that of VP37 (Figs. 7, 8, and 9).

<Reviewer #2>

Authors study in detail the existence and possible function of the putative VP53 protein of the fabavirus Broad bean wilt virus 2 (BBWV2). They engineer several recombinant and mutant constructs in order to decipher the involvement of this protein in the viral infection. They provide evidence of the possible mechanism of translation initiation and show that the protein is essential for the life cycle of BBWV2. I think they present very interesting and powerful results to support their conclusions, like the ones obtained with the natural compensatory mutations found after removal of the start codon of VP53 or the experiments in which VP53 is expressed independently after VP37.

The part that I have found less clear is the one about the leaky expression of the proteins. And this is probably because of two things. The first one is that it is not clear to me what is the starting hypothesis of the authors in this regard. Do they thought BBWV2 would behave like their comovirus cousins, using a dual mechanism for the expression of the VP proteins or do they thought that leaky expression would be the most likely scenario for BBWV2? I think the explanation of the results vary depending if it is one or the other.

→ Thank you for the comments. So far, it has been hypothesized that BBWV2 RNA2 encodes two largely overlapping polyproteins using two alternative in-frame initiation codons (at nucleotide positions 236 and 620). However, this hypothesis is based on the similarities to other related viruses in the family *Secoviridae* and has not yet been experimentally demonstrated. In this study, we demonstrated that VP53 is truly expressed during viral replication and essential for successful infection of BBWV2. For clarification, we have revised our MS accordingly (Lines 79-81 and 98-101)

The second one I think is because of the experiments performed *in silico* with the AUG surrounding regions. These results are presented in Figure 4C-F, but they are not mentioned in the discussion. Thus, I do not know if they actually match the leaky expression hypothesis or not. Besides, in the *in silico* analyses of the start initiation, authors state that “nucleotide sequences around both start codons were highly conserved among BBWV2 isolates”. I can see this for VP37, but not in the case of VP53, at least not in the part upstream the AUG, where only 4 out of 10 nucleotides are conserved. Then authors say that the sequences around AUG are unfavorable for eukaryotic ribosome recognition, and that this is surprising. However, to my knowledge, authors do not provide any possible explanation for this in the results or in the discussion. Authors also perform secondary structure analysis in these regions and they found hairpin RNA structures they consider intriguing. Again, do these data fit the leaky expression hypothesis or not? This is not clearly explained either in my opinion.

→ Thank you for the important comments. We agree with the reviewer’s comments regarding the insufficient explanation provided. Thus, we have revised our MS to provide a more detailed explanation for how the sequence contexts and RNA structures surrounding the start codons for VP53 and VP73 could function in regulating translation efficiency and ribosomal leaky scanning (Lines 611-639).

In stating “nucleotide sequences around both start codons were highly conserved among BBWV2 isolates”, our aim was to elucidate the conservation of the sequence contexts at the most important positions -4, -3, -2, -1, and +4 surrounding the translation initiation codon. However, for clarification, we have revised our MS accordingly (Lines 370-372).

Besides this, there are other minor comments or concerns that I address below:

- I think when authors state in lines 330-331 that “Our analysis clearly detected both VP53-Flag and VP37-Flag at the expected size” they are too categorical. They observe bands at the anticipated size of VP53-Flag and VP37-Flag. However, for example the band matching the size of VP53-Flag could be VP37-Flag with some posttranslational modifications, or it could be something else. Unlikely, but possible. Mass spectrometry could be a good complement to try to identify N-terminal peptides of VP53.

→ Thank you for the very helpful comments. As per the suggestion, we performed LC-MS/MS analysis on a gel fragment corresponding to the molecular weight of the VP53 protein (~53 kDa). This analysis demonstrated that VP53 is indeed expressed during BBWV2 infection (Lines 355-362 and Supplementary Fig. S2).

- Probably Figs. 1, 2 and 3 or at least Figure 1 and 2 could be merged together.

→ As per the suggestion, Figs. 1 and 2 were merged together (Fig. 1)

- In Fig. 2C, why there is no Aniline blue signal in the healthy plant?

→ The healthy plant was included as a negative control for aniline blue staining. The samples stained with aniline blue were indicated on the right side of the images (Fig. 1D).

- In Fig. 4B, Fig. 6C and D and Fig. 10E, I think the labels VP37 and VP53 should be VP37-FLAG and VP53-FLAG.

→ Corrected.

- In Fig. 7 it is not clear to me to which sequence corresponds the electropherogram that appears on top of the alignments.

→ The electropherogram is for Progeny #1, shown as a representative. We have revised the figure to clearly indicate it (Fig. 6).

- In Fig 9B I am not able to see any differences between infected and not-infected plants. Probably it could be a good option to move these images to supplementary and include Fig 10A and 10B in this Figure 9 instead.

→ As per the suggestion, Fig. 9B was move to supplementary and enlarged to clearly show the symptoms in the infected plants (Supplementary Fig. S5)

- Line 107: VP53 is essential...

→ Corrected.

- Line 111: remove the “;” after pBBWB2-R2-OE

→ Corrected.

- Why a WB is not performed using anti-GFP, as in Fig 8E, to verify the presence of VP53 instead of making specific constructs with Flag? Wouldn't it be more informative to have the data of the presence of VP53 in samples already analyze by confocal microscopy?

→ When we planned this study, we decided to generate pBBWV2-R2-53/37-GFP to use it in visual tracking of the virus and pBBWV2-R2-53/37-Flag for Western blot detection of VP53 and VP37. And then, the experiments utilizing pBBWV2-R2-53/37-GFP and its derivatives, as well as pBBWV2-R2-53/37-Flag and its derivatives, were performed almost simultaneously. We ask for your understanding of our experimental designs.

- In lines 340-343, the word “sequence” appears too many times.

→ Corrected (Lines 370-373).

- The fact that GFP exacerbated the symptoms and Flag did not is not discussed. Do the authors have any hypothesis to explain this?

→ Thank you for the comments. We have revised our MS to provide a possible explanation for why tagging with GFP at the C-terminus of VP53/VP37, but not with Flag, resulted in an enhancement of viral symptom severity (Lines 679-694).

<Reviewer #3>

The study analyzed the protein VP53 potentially encoded by Broad bean wilt virus 2 (BBWV2) using genetic, virological, cell biology, and molecular biology approaches to investigate the functionality and expression of VP53. Overall, the research provides new data for a deeper understanding of BBWV2 pathogenesis. I have a few suggestions to further improve the quality of this manuscript:

1. To confirm that VP53 is indeed encoded and expressed by BBWV2, it is recommended to use a VP53-specific antibody (e.g., N-terminal specific antibody against VP53) for further validation of VP53 expression.

→ Thank you for the critical comments. To confirm the expression of VP53, we performed an additional LC-MS/MS analysis on a gel fragment corresponding to the molecular weight of the VP53 protein (~53 kDa). This analysis demonstrated that VP53 is indeed expressed during BBWV2 infection (Lines 355-362 and Supplementary Fig. S2).

In addition, it is noteworthy that the expression of VP53 was detected under natural virus infection conditions in the upper systemic leaves, independent of the 35S promoter (Figs. 3B and 5D). The methodology for *in vivo* production of infectious viral RNAs using the 35S promoter and binary vectors upon *Agrobacterium*-mediated inoculation has been widely used to study the molecular biology of plant RNA viruses (Mori et al., 1991, *J Gen Virol* 72:243-246; Boyer et al., 1994, *Virology* 198:415-426). In this methodology, the 35 promoter serves to generate initial viral RNA transcripts in inoculated plant leaves. The initial viral RNA transcripts are then translated and utilized as templates to initiate virus replication. Subsequently, the virus replicates under natural virus infection conditions, particularly in upper systemically infected leaves. In addition, the tagging with GFP or Flag at the C-terminus of VP53/37 did not affect viral RNA replication levels (Lines 302-303 and Supplementary Fig. S1). As shown in Figs. 3B and 5D, we detected the accumulation of VP53 in the upper systemic leaves infected with BBWV2-53/37-Flag, providing evidence that VP53 is expressed during BBWV2 infection under physiological conditions (natural virus infection conditions) independent of the 35S promoter. For clarification, we have revised our MS accordingly (Lines 594-603).

2. line 329, "N. benthamiana" should be italicized. Check the entire manuscript for similar errors.

→ Corrected.

3. line 353, I wonder whether the hairpin RNA structures affect the translation of both VP53 and VP37.

→ Thank you for the comments. Characterizing the functionality of RNA structures requires substantial experimentation. Therefore, we are currently performing a separate study to further characterize the functional significance of the stem-loop RNA structures, which

were predicted near the AUG start codons for VP53 and VP37, in regulating the translation efficiency of VP53 and VP37. We hope to show our findings in a publication in near future. In this study, we have revised our MS to provide a more detailed explanation for how the sequence contexts and RNA structures surrounding the start codons for VP53 and VP73 could function in regulating translation efficiency and ribosomal leaky scanning (Lines 611-639).

4. line 383, avoid stating "data not shown." Provide the relevant data to support the claims made in the paper.

→ We have revised to provide the data in supplementary (Line 414 and Supplementary Fig. S4)

5. line 426, since the mentioned mutations did not completely block VP53 expression, I suggested introducing more stop codons at the N-terminus of VP53 and examine their impact on BBWV2 infection.

→ In bean pod mottle virus (BPMV; genus *Comovirus*, family *Secoviridae*), a mutation at the start codon for P58 (a counterpart of VP53) resulted in spontaneous second-site mutations at different positions that restore the translation of P58 (Lin et al., 2014, J Virol 88:3213-3222). In addition, our results suggest that VP53 is essential for the replication of BBWV2 RNAs as well as viral systemic infection (Fig. 7). Therefore, it may be difficult to examine the function of VP53 simply by completely blocking VP53 expression through the introduction of additional stop codons at the N-terminus of VP53. As an alternative approach, we tested the BBWV2 mutants containing the deletion of the N-terminus of VP53 (i.e. BBWV2-R2- Δ 53N1-GFP and pBBWV2-R2- Δ 53N2-GFP) (Fig. 7). In addition, we tested a series of BBWV2 recombinants RNA2 constructs, which contain the deletion of the N-terminus of VP53 and are capable of expressing either the full-length VP53 or its C-terminal deletions from the gene insertion cassette (Fig. 8). Based on this approach, we demonstrated that the N-terminal region of VP53, which does not overlap with VP37, is sufficient for its functionality (Fig. 8 and Supplementary Fig. S5).

6. line 471, I wonder whether the deletion of the N-terminus of VP53 (Fig. 8A) affects the RNA structure, thereby influencing BBWV2 infection.

→ In pBBWV2-R2- Δ 53N1-GFP, although the genomic region (nucleotide positions 242-578) was deleted, the formation of the stem-loop RNA structures near the start codon for VP37 were still predicted, because the deletion mutant contained the sequences required for the stem-loop RNA structures (nucleotide positions 579-659) (Figs. 3E and 7A). Furthermore, as shown in Fig.9C, the deletion of the genomic region corresponding to the N-terminus of VP53 (nucleotide positions 242-578) did not diminish the expression of VP37 because BBWV2- Δ 53N1-F-VP53:Flag systemically infected the inoculated plants and highly accumulated VP37 as well as VP53 (Fig. 9).

Merge Fig. 1 and Fig. 2 for better presentation.

→ As per the suggestion, Figs. 1 and 2 were merged together (Fig. 1).

7. In Fig. 6D, explain why the expression of VP53 in BBWV2-53^{ATG236TAA}/37-Flag is higher than in BBWV2-53/37-Flag, it seems that the mutation enhances VP53 expression.

→ As shown in Fig. 6B, the progenies of BBWV2-53^{ATG236TAA}/37-Flag acquired a new G-to-A substitution at position 257, leading to the expression of an N-terminally 7-amino-acid-truncated form of VP53. It is possible that the translation efficiency of the sequence context of the new start codon could differ from that of the original start codon for VP53. We have revised our MS to provide explanation regarding this point (Lines 703-709).

8. For Fig. 7, consider using RNA-seq (deep sequencing) to further characterize the mutations of BBWV2-53 mutants during infection.

→ RNA viruses exist as quasispecies. Therefore, analyzing the consensus sequences of a virus progeny population is beneficial for understanding the representative characteristics of the virus. Although we reexamined carefully the electropherogram data obtained for the sequencing results of the progeny viruses of BBWV2-53^{G238A}/37-GFP, BBWV2-53^{ATG236TAA}/37-GFP, or BBWV2-53^{ATG236TAA}/37-Flag, no additional mutations creating new start codons for VP53 were found.

9. I wonder how about the mutation of residue 257 to a stop codon on BBWV2 infection.

→ As explained above, it may be difficult to examine the function of VP53 simply by completely blocking VP53 expression through the introduction of additional stop codons at the N-terminus of VP53. As an alternative approach, we tested the BBWV2 mutants containing the deletion of the N-terminus of VP53 (i.e. BBWV2-R2-Δ53N1-GFP and pBBWV2-R2-Δ53N2-GFP) (Fig. 7). In addition, we tested a series of BBWV2 recombinants RNA2 constructs, which contain the deletion of the N-terminus of VP53 and are capable of expressing either the full-length VP53 or its C-terminal deletions from the gene insertion cassette (Fig. 8). Based on this approach, we demonstrated that the N-terminal region of VP53, which does not overlap with VP37, is sufficient for its functionality (Fig. 8 and Supplementary Fig. S5).

10. Fig. 9B, whether these mutations affect the symptom severity of BBWV2? and molecular detection should be conducted to verify the symptom phenotype shown in Fig. 9B.

→ Fig. 9B was moved to supplementary and enlarged to clearly show the symptoms in the infected plants (Supplementary Fig. S5). BBWV2-Δ53N1-VP53, BBWV2-Δ53N1-VP53ΔC1, and BBWV2-Δ53N1-VP53ΔC2 induced more severe symptoms (i.e. leaf size reduction, stunting, and vein yellowing) than wt BBWV2 and BBWV2-Δ53N1-VP53ΔC3 in *N. benthamiana* (Lines 521-524). In addition, as shown in Fig. 8B and C, we already performed RT-qPCR analysis to examine the accumulation levels of the BBWV2 recombinant viruses in systemically infected leaves. The accumulation levels of both RNA1 and RNA2 of BBWV2-Δ53N1-VP53, BBWV2-Δ53N1-VP53ΔC1, and BBWV2-Δ53N1-VP53ΔC2 were significantly higher compared to those of wt BBWV2 and BBWV2-Δ53N1-VP53ΔC3 (Fig. 8B and C). The elevated accumulation of viral RNAs in BBWV2-Δ53N1-VP53, BBWV2-Δ53N1-VP53ΔC1, and BBWV2-Δ53N1-

VP53 Δ C2 could be associated with their increased symptom severity. For clarification, we have revised our MS accordingly (Lines 521-524 and 720).

REVIEWERS' COMMENTS:

Reviewer #1 (Remarks to the Author):

The authors addressed my concerns carefully. I am sorry for the typo of VP53 to VP57.

Reviewer #2 (Remarks to the Author):

I think authors did a very good job answering my previous concerns. They used mass spectrometry for the detection of VP53 as I suggested and they modified the manuscript to clarify things I thought were not clear enough in the previous submission. Still, in my opinion, in order for the article to be ready for publication a few minor changes must be done:

- I might be mistaken, but I cannot see the change in Fig. 6, that the authors mentioned regarding clarification of the electropherogram.
- In figures 4, 7 and 8 authors used t-test to compare more than two samples. I think an ANOVA would be more suitable statistically for the comparison of more than two samples.
- Line 598, an "S" is missing after "35".

Reviewer #3 (Remarks to the Author):

The authors addressed my concerns in a satisfied manner.

Reviewers' comments:

<Reviewer #1>

The authors addressed my concerns carefully. I am sorry for the typo of VP53 to VP57.

→ Thanks a lot.

<Reviewer #2>

I think authors did a very good job answering my previous concerns. They used mass spectrometry for the detection of VP53 as I suggested and they modified the manuscript to clarify things I thought were not clear enough in the previous submission. Still, in my opinion, in order for the article to be ready for publication a few minor changes must be done:

- I might be mistaken, but I cannot see the change in Fig. 6, that the authors mentioned regarding clarification of the electropherogram.

→ The electropherograms is for Progeny #1. For clarification, we indicated it using red line (Fig. 6).

- In figures 4, 7 and 8 authors used t-test to compare more than two samples. I think an ANOVA would be more suitable statistically for the comparison of more than two samples.

→ Thank you for the suggestion. We have reanalyzed our data using one-way ANOVA with Tukey's HSD test ($P < 0.05$) and revised our manuscript (Figs. 4C, 4D, 7C, 7D, 8B, and 8C; Lines 739-740, 987-988, 1052-1054, and 1088-1090)

- Line 598, an "S" is missing after "35".

→ Corrected (Line 423).

<Reviewer #3>

The authors addressed my concerns in a satisfied manner.

→ Thanks a lot.